# Multi-conditioned Graph Diffusion for Neural Architecture Search

**Rohan Asthana**                                          *rohan.asthana@fau.de*
*Friedrich-Alexander-Universität Erlangen-Nürnberg*
*Erlangen, Germany*

**Joschua Conrad**                                          *joschua.conrad@uni-ulm.de*
*Universität Ulm*
*Ulm, Germany*

**Youssef Dawoud**                                          *youssef.dawoud@fau.de*
*Friedrich-Alexander-Universität Erlangen-Nürnberg*
*Erlangen, Germany*

**Maurits Ortmanns**                                          *maurits.ortmanns@uni-ulm.de*
*Universität Ulm*
*Ulm, Germany*

**Vasileios Belagiannis**                                          *vasileios.belagiannis@fau.de*
*Friedrich-Alexander-Universität Erlangen-Nürnberg*
*Erlangen, Germany*

**Reviewed on OpenReview:** *https://openreview.net/forum?id=5VotySkajV*

## Abstract

Neural architecture search automates the design of neural network architectures usually by exploring a large and thus complex architecture search space. To advance the architecture search, we present a graph diffusion-based NAS approach that uses discrete conditional graph diffusion processes to generate high-performing neural network architectures. We then propose a multi-conditioned classifier-free guidance approach applied to graph diffusion networks to jointly impose constraints such as high accuracy and low hardware latency. Unlike the related work, our method is completely differentiable and requires only a single model training. In our evaluations, we show promising results on six standard benchmarks, yielding novel and unique architectures at a fast speed, i.e. less than 0.2 seconds per architecture. Furthermore, we demonstrate the generalisability and efficiency of our method through experiments on ImageNet dataset.

## 1 Introduction

The design of neural network architectures has been normally a manual and time-consuming task, requiring domain expertise and trial-and-error experimentation (Elsken et al., 2019). Neural Architecture Search (NAS) addresses this limitation by leveraging data-driven methods to automatically search for well-performing neural network architectures (Liu et al., 2019; Howard et al., 2019; Pham et al., 2018). Existing works in NAS mostly represent the architectures as graphs and include search based methods (Li & Talwalkar, 2020; White et al., 2021b), reinforcement learning (Zoph & Le, 2017; Tian et al., 2020), and evolution-based

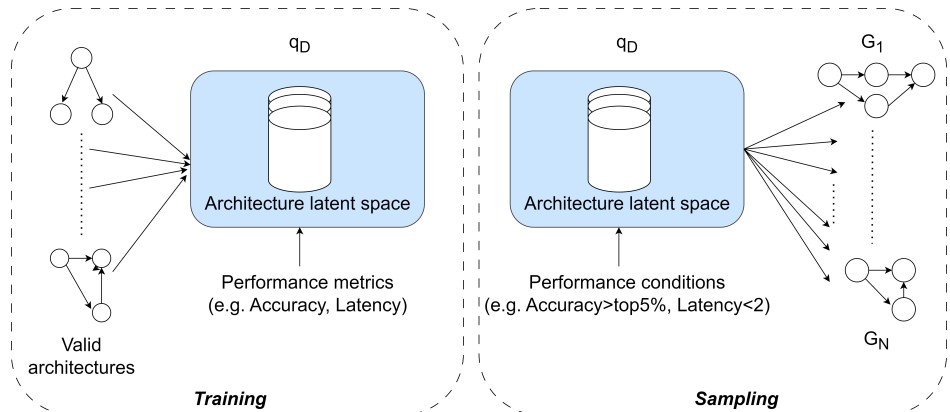

Figure 1: Overview of our approach. We train a discrete graph diffusion model (denoted as $q_D$) on valid architectures from the architecture search space along with their performance metrics (eg. accuracy, latency). After training, we sample architectures given the required conditions (eg. accuracy>top5%, latency<2).

approaches (Real et al., 2019; Chu et al., 2020). However, the large size of the architecture search space makes it challenging for these methods to search for high-performing topologies.

To accelerate the architecture search, generative methods reduce the search queries by learning the architecture search space and optimising the latent space from which a generator network draws architectures (Rezaei et al., 2021; Huang & Chu, 2021). These methods not only enhance the efficiency but also capture intricate architecture distributions, generating novel architectures. However, the choice of graph generative model significantly impacts the NAS search time. The existing methods employ complex GAN-based generators (Rezaei et al., 2021) or use computationally intensive supernets (Huang & Chu, 2021). More recently, An et al. (2023) employs conditional diffusion processes guided by a classifier while Lukasik et al. (2022) uses a simple generator paired with a surrogate predictor. However, these methods require separate predictor networks for the generated architecture performance. Hence, we present a diffusion-based generative approach that is completely differentiable and thus training involves only a single model. As a result, we reach promising performance with much smaller search time.

Denoising diffusion probabilistic models (DDPMs) (Ho et al., 2020) have recently gained attention because of their ability to effectively model complex data distributions through an iterative denoising process. DDPMs offer precise generative control, improving distribution coverage compared to other generative models (Dhariwal & Nichol, 2021). This characteristic makes diffusion models particularly appealing for NAS, as they fulfil the requirement to generate neural network architectures and eventually facilitate the exploration of the search space. In addition to their superior performance, diffusion models excel in conditional generation through the classifier-free guidance technique (Ho & Salimans, 2021). This technique enables the conditioning of diffusion models on a specific target class, allowing the model to generate samples belonging to that class without utilizing the gradients of an external classifier. Previous studies show that along with image synthesis, classifier-free guidance works well in molecule (graph) synthesis using graph diffusion networks (Hoogeboom et al., 2022). Moreover, Giambi & Lisanti (2023) demonstrated the capability of classifier-free guidance approach using multiple conditions. Motivated by this idea, we present a multi-conditioned graph diffusion model in which constraints such as high model accuracy and low latency jointly contribute to architecture sampling.

We introduce a graph diffusion-based NAS approach (DiNAS), depicted in Figure 1, that utilises discrete conditional graph diffusion processes to generate high-performing neural network architectures [1]. We leverage classifier-free (CF) guidance, initially developed for image tasks, and extend its application to graph models. Additionally, to impose multiple constraints, we utilize a multi-conditioned CF guidance technique, and apply it within our graph diffusion framework. To demonstrate the effectiveness of our proposed method, we perform extensive evaluations on six standard benchmarks, including experiments on ImageNet (Deng

---

[1]The code for our paper is available at `https://github.com/rohanasthana/DiNAS`.

et al., 2009), and ablation studies to demonstrate state-of-the-art performance and faster generation rate (less than 0.2 seconds per architecture on a single GPU) compared to the prior work. To the best of our knowledge, this is the first formulation of NAS using multi-conditioned graph-based diffusion models. In summary, we claim that guided graph diffusion, specifically discrete graph diffusion with multi-conditioned classifier-free-guidance-based NAS approach, should work better than previous generative and traditional NAS methods due to the model's ability to perform architecture generation in a controlled guided manner without the need of an external predictor. Our claims are supported by empirical evidence detailed in Section 5. Our contributions are as follows:

- We introduce a differentiable generative NAS method, which employs discrete conditional diffusion processes to learn the architecture latent space by training a single model.

- We propose a multi-conditioned diffusion guidance technique for graph diffusion networks, effectively applied within our NAS approach.

- We demonstrate promising results in six standard benchmarks while using less or same number of queries with rapid generation of novel and unique high-performing architectures.

## 2 Related Work

**Neural Architecture Search (NAS)** Automating the neural network architectural design has gained substantial interest in the past few years (Liu et al., 2019; Jin et al., 2019; Zoph et al., 2018; Bender et al., 2018; Shala et al., 2023). A straightforward approach is to randomly select and evaluate architectures from the search space (Li & Talwalkar, 2020). However, the lack of optimisation in the search space makes this approach inefficient. To address this limitation, earlier works rely on reinforcement learning (Zoph & Le, 2017; Baker et al., 2017; Franke et al., 2021) to discover well-performing architectures. Gradient-based approaches (Brock et al., 2018; Chen et al., 2021b; Yang et al., 2020) employ gradient-based optimisation, while evolutionary methods (Real et al., 2019; 2017) deploy evolutionary algorithms to perform the search. Although these approaches exhibit faster search pace than random search due to the optimisation, they are still regarded slow in searching high-performing architectures (Liu et al., 2019). Another major challenge with search-based methods is the requirement to train networks at each iteration (Luo et al., 2022). This becomes particularly problematic when NAS approaches require a substantial number of iterations to generate well-performing architectures, which is often the case with reinforcement learning-based methods. This issue is solved by the recently developed generative methods (Lukasik et al., 2021; Rezaei et al., 2021; Lukasik et al., 2022; An et al., 2023), which reduce the search time by learning the architecture search space. The generative NAS method by Lukasik et al. (2022) utilises a generator, a surrogate predictor, paired with a latent space optimisation technique to generate high performing architectures represented as graphs. This generation is performed node by node, which requires multiple passes of a graph neural network, with each pass to generate a node. In contrast, our approach learns and generates all the nodes and the edges of the whole graph (as a representation of architectures) together using a diffusion model, effectively reducing the generation time. While the method proposed by An et al. (2023) also utilises diffusion models for neural architecture synthesis, there are several key differences to note. First, unlike their approach, we employ discrete graph diffusion instead of continuous graph diffusion. This allows our model to retain the structural information and sparsity of the graphs (Vignac et al., 2023). Second, we eliminate the need of an external predictor to predict the accuracies from noisy data, as done by An et al. (2023), which allows our model to omit the dependency on noisy data classification (Ho & Salimans, 2021). Following the same direction, we present a generative model that reduces the search time compared to the prior work, while minimising the performance loss.

**Diffusion Models** Although the original idea of data generation through diffusion goes back several years (Sohl-Dickstein et al., 2015), diffusion models later gained popularity for image (Ho et al., 2020; Rombach et al., 2022; Saharia et al., 2022)), text (Austin et al., 2021) and more recently graph generation (Wang et al., 2022). The ability of diffusion models to effectively synthesise graphs motivates our work to formulate NAS as a graph generation problem. Nevertheless, the generation of well-performing architectures requires

conditional generation through guidance, e.g. by specifying the minimum architecture accuracy. Hence we utilize a formulation of classifier-free guidance for graph-diffusion networks. Then, we introduce a multi-conditioned graph-diffusion approach that accounts for several constraints in the architecture generation.

## 3   Background

### 3.1   Denoising Diffusion Probabilistic Models

Denoising Diffusion Probabilistic Models (DDPMs) (Ho et al., 2020) comprise two fundamental processes, namely, forward and reverse processes. The forward process sequentially corrupts the data sample $\mathbf{x}$ using a noise model $q$ that follows a Gaussian distribution until $\mathbf{x}$ reaches a state of pure noise. The noisy variants of $\mathbf{x}$ are denoted as $(\mathbf{x^1}, \mathbf{x^2}, \ldots, \mathbf{x^T})$, where $T$ represents the total number of corruption steps. Subsequently, the reverse process involves learning a denoising model represented as a deep neural network $\phi_\theta$ with parameters $\theta$ to estimate the noise state of sample $\mathbf{x}$ at time step $t-1$, i.e. $\mathbf{x}^{t-1}$ given the current state $\mathbf{x}^t$. This is achieved using a scoring function that maximises the likelihood of $\mathbf{x}^{t-1}$. Formally, the scoring function is defined as $S_F = \nabla_{\mathbf{x}^{t-1}} \log p_\theta(\mathbf{x}^{t-1}|\mathbf{x}^t)$ which corresponds to the gradient of the log-likelihood with respect to state $\mathbf{x}^{t-1}$. Following the network training, another data point can be sampled from a noisy prior (denoted as $\mathbf{z}^T$), and by iterative denoising the data point (i.e. predicting $\mathbf{z}^{t-1}$ from $\mathbf{z}^t$), a sample $\mathbf{z}^0$ is obtained, which corresponds to the original data distribution. This process is referred to as the sampling process.

Diffusion models can generate high-quality samples from complex data distributions. However, in our task, we do not intend to sample from the entire distribution but a subset of it containing high-performing and/or low latency architectures. Hence, diffusion models need to be modified to incorporate conditioning. This can be achieved using conditional diffusion models.

### 3.2   Conditional diffusion models with guidance

The conditional diffusion model estimates the distribution $p_\theta(\mathbf{x}^{t-1}|\mathbf{x}^t, y)$. From Bayes rule, we have:

$$p_\theta(\mathbf{x}^{t-1}|\mathbf{x}^t, y) \propto p_\theta(\mathbf{x}^t, y|\mathbf{x}^{t-1})p(\mathbf{x^{t-1}}). \tag{1}$$

To ensure the balance between sampling diversity and quality, generative models can incorporate guidance scale $\gamma$, modifying the Eq. 1 to:

$$p_\theta(\mathbf{x}^{t-1}|\mathbf{x}^t, y) \propto p_\theta(\mathbf{x}^t, y|\mathbf{x}^{t-1})^\gamma p(\mathbf{x^{t-1}}). \tag{2}$$

Specifically, increasing $\gamma$ sharpens the distribution which favors enhanced sample quality at the expense of sample diversity during the sampling process, referred to as guidance in diffusion models. To guide a diffusion model for labelled data, the model is conditioned on the classification target $y$ and the score function $\nabla_{\mathbf{x}^{t-1}} \log p_\theta(\mathbf{x}^{t-1}|\mathbf{x}^t, y)$ is computed. Dhariwal & Nichol (2021) approach this problem using an external classifier (parameterised by $\psi$) where the score function $S_F$ is modified to include the gradients of the classifier. The reformulated score function is then the weighted sum of the unconditional score function and the conditioning term obtained by the classifier, defined as:

$$\nabla_{\mathbf{x}^{t-1}} \log p_\theta(\mathbf{x}^{t-1}|\mathbf{x}^t, y) = \nabla_{\mathbf{x}^{t-1}} \log p_\theta(\mathbf{x}^{t-1}|\mathbf{x}^t) + \gamma \nabla_{\mathbf{x}^{t-1}} \log p_\psi(y|\mathbf{x}^{t-1}). \tag{3}$$

While we have successfully expressed the reverse denoising process as a weighted sum of two score functions, estimation of the conditional score function requires training a separate classifier. Moreover, calculating $\log p_\psi(y|\mathbf{x}^{t-1})$ requires inferring $y$ from noisy data $\mathbf{x}^t$. Although feeding noisy data to the classifier yields decent performance, it disrupts the robustness of the model since it ignores most of the original input signal. To address this issue, Ho & Salimans (2021) came up with the classifier-free guidance, which develops the classifier using the generative model itself. In this case, the score function is defined as:

$$\nabla_{\mathbf{x}^{t-1}} \log p_{\theta_\gamma}(\mathbf{x}^{t-1}|\mathbf{x}^t, y) = (1-\gamma)\nabla_{\mathbf{x}^{t-1}} \log p_\theta(\mathbf{x}^{t-1}|\mathbf{x}^t) + \gamma \nabla_{\mathbf{x}^{t-1}} \log p_\theta(\mathbf{x}^{t-1}|\mathbf{x}^t, y). \tag{4}$$

Eq. 4 demonstrates that it is possible to achieve the same behaviour as the classifier-based guidance without explicitly using a classifier. This is achieved through a weighted sum, specifically a barycentric combination, of the conditional and unconditional score functions.

### 3.3 Discrete Graph Diffusion

Diffusion models typically work in a continuous space and apply Gaussian noise to the data (Ho et al., 2020; Saharia et al., 2022). Training a diffusion model to generate graphs in the same manner, however, leads to the loss of graph sparsity and structural information. DiGress, a discrete diffusion approach proposed by Vignac et al. (2023), addresses this problem with a Markov processes as discrete noise model. In this case, the graph comprises of nodes and edges, both being categorical variables, and the goal is to progressively add or remove edges as well as change graph node categories. Hence, the diffusion process is applied on the node categories $\mathbf{X}$ and edges $\mathbf{E}$. Eventually, this model solves a simple classification task for nodes and edges instead of a complex distribution learning task, normally performed in generative models like VAEs (Kingma & Welling, 2013) or DDPMs (Ho et al., 2020). Our approach, in principle, follows DiGress to generate graphs which correspond to neural network architectures.

At each forward step, discrete marginal noise is added to both $\mathbf{X}$ and $\mathbf{E}$ using the transition probability matrices $Q_X$ and $Q_E$ respectively, which incorporate the marginal distributions $m'_X$ and $m'_E$. We select the noisy prior distribution such that it is close to the original data distribution. Then, the transition matrices are defined as follows:

$$Q_X^t = \bar{a}^t I + (1 - \bar{a}^t) 1_i m'_X; \quad Q_E^t = \bar{a}^t I + (1 - \bar{a}^t) 1_j m'_E, \tag{5}$$

where $I$ is the identity matrix, $1_i$ and $1_j$ are the indicator functions, $t$ is the time-step, and $\bar{a}^t$ is the cosine schedule defined as $\bar{a}^t = \cos(0.5\pi(t/T + s)/(1 + s))^2$ with $s$ close to 0.

**Training** For the reverse (denoising) step, a Graph Transformer network $\phi_\theta$, parameterised by $\theta$, is employed. This network learns the mapping between the noisy graphs $\mathbf{G^t}$ and the corresponding clean graphs $\mathbf{G}$. During training, $\phi_\theta$ can take noisy graphs at any time step $t \in (1, .., T)$ to predict the clean graph. The loss functions for $\mathbf{X}$ and $\mathbf{E}$ are based on the cross-entropy between their respective predicted probabilities $\hat{p}^G = (\hat{p}^X, \hat{p}^E)$ and the ground-truth graph $\mathbf{G} = (\mathbf{X}, \mathbf{E})$. The total loss is then, a weighted sum of node-level and edge-level losses, which is given by:

$$L_G(\hat{p}^X, \mathbf{X}, \hat{p}^E, \mathbf{E}) = \sum_{1 \leq i \leq n} CE(x_i, \hat{p}_i^X) + \lambda \sum_{1 \leq i,j \leq n} CE(e_{ij}, \hat{p}_{ij}^E), \tag{6}$$

where $CE$ is the cross-entropy loss function, $\lambda$ is a parameter to weight the importance of nodes and edges and, n is the number of nodes.

**Sampling** Let the posterior distribution be $p_\theta$. We start from a noisy prior distribution $\mathbf{G}^T \sim (q_X(n_T) \times q_E(n_T))$, where $n_T$ is sampled from the node distribution in the training data. Then, we estimate the node and edge distributions $p_\theta(x_i^{t-1}|\mathbf{G}^t)$ and $p_\theta(e_{ij}^{t-1}|\mathbf{G}^t)$ using the predicted probabilities $\hat{p}_i^X$ and $\hat{p}_{ij}^E$. This can be written as:

$$p_\theta(x_i^{t-1}|\mathbf{G}^t) = \sum_{x \in \mathbf{X}} p_\theta(x_i^{t-1}|x_i = x, \mathbf{G}^t)\hat{p}_i^X(x); \quad p_\theta(e_{ij}^{t-1}|\mathbf{G}^t) = \sum_{e \in \mathbf{E}} p_\theta(e_{ij}^{t-1}|eij = e, \mathbf{G}^t)\hat{p}_{ij}^E(e). \tag{7}$$

Finally, sampling new graphs can be seen as iteratively estimating the distribution $p_\theta(\mathbf{G}^{t-1}|\mathbf{G}^t)$ until a clean graph $\mathbf{G}^0$ is obtained. $p_\theta(\mathbf{G}^{t-1}|\mathbf{G}^t)$ can be seen as the product of the node and edge distributions marginalised over predictions from the network $\phi_\theta$:

$$p_\theta(\mathbf{G}^{t-1}|\mathbf{G}^t) = \prod_{1 \leq i \leq n} p_\theta(x_i^{t-1}|\mathbf{G}^t) \prod_{1 \leq i,j \leq n} p_\theta(e_{ij}^{t-1}|\mathbf{G}^t). \tag{8}$$

The above model successfully handles sparse categorical graph data in a discrete manner, synthesising graphs from complex data distributions. Therefore, it is suitable for our problem. Nevertheless, in our task, we seek to introduce conditioning in discrete graph diffusion models through classifier-free (CF) guidance. To that end, we propose next a multi-conditioned graph diffusion formulation for NAS.

# 4 Method

Consider the diffusion model $q_D$ comprising of a neural network $\phi_\theta$ paramterised by $\theta$. During training, the model $q_D$ takes the directed acyclic graph $\mathbf{G}$ as input and learns to reconstruct $\mathbf{G}$ from the noisy version $\mathbf{G^t}$, where $t \in (1, \ldots, T)$ is the number of diffusion time steps. This reconstruction is essentially performed by learning to estimate the actual data distribution $\mathcal{G}$ from the noisy version of $\mathcal{G}$, which we denote as $P_N$, through iterative denoising. Following the training of $\phi_\theta$, we aim to generate DAGs representing high-performing neural network architectures using samples from $P_N$, where we denote a sample as $\mathbf{z}$.

Our directed acyclic graph (DAG) representation of architectures follows the standard cell-based NAS search spaces (Liu et al., 2019; Klyuchnikov et al., 2020), where each cell is a DAG. $\mathbf{G}$ consists of a set of nodes and edges. The sequence of nodes in $\mathbf{G}$ is represented by $\mathbf{X} = [v_1, v_2 \ldots v_n]$, where the number of nodes is $n$, and the edges as the adjacency matrix $\mathbf{E}$ of shape $(n, n)$. Hence, each DAG is represented by $\mathbf{G} = (\mathbf{X}, \mathbf{E})$. Each node is a categorical variable, describing operations, e.g., 1x1 convolution, while each edge is a binary variable, specifying the presence or absence of the connection between nodes. In addition, $\mathbf{G}$ maps to the ground-truth performance metrics $P$ e.g., the accuracy and latency of each DAG.

Our objective is twofold, namely to generate valid cells $C_v = (\mathbf{X_v}, \mathbf{E_v})$ from the latent variable $\mathbf{z}$, sampled from the noise distribution $P_N$ and, second, to learn the mapping between the valid cell $C_v$ and its corresponding performance metrics $P$. The learned mapping is then used to generate high-performing cells with accuracy close to the maximum achievable accuracy or cells with latency below a certain latency constraint. Note that a cell is valid when the corresponding DAG is connected and includes a realistic sequence of nodes.

## 4.1 Diffusion based NAS

We consider the unconditional and conditional graph generation. First, we present the unconditional model that learns to generate valid cells. Since some of the generated cells might have poor performance, we propose the single conditioned and multi-conditioned graph diffusion models to generate just the high-performing cells based on metrics like the model accuracy and latency.

### 4.1.1 Unconditional model

Our unconditional model is based on the discrete denoising graph diffusion model (Vignac et al., 2023), outlined in Section 3.3. The forward process involves adding discrete marginal noise $Q_X^t$ and $Q_E^t$ (Eq. 5) to both nodes $\mathbf{X}$ and edges $\mathbf{E}$ respectively. To perform denoising, we employ the Graph Transformer network $\phi_\theta$, which is trained to predict clean graphs $\mathbf{G}$ from noisy graphs $\mathbf{G^t}$. While this model effectively captures the data distribution for undirected graphs, it lacks the ability to incorporate directional information of DAGs. This directional information depicts the flow of data from input to output in the cells and hence is crucial for generating valid cells. To address this limitation, we integrate into our model the positional encoding technique by Vaswani et al. (2017b). In detail, we add sinusoidal signals of different frequencies to the node features $\mathbf{X}$ before passing them through the Graph Transformer $\phi_\theta$, thereby enhancing the network's capability to consider sequential information.

Despite the ability of our unconditional model in forming valid cells necessary for complete network architectures, our goal lies in generating a particular subset of the learned architecture distribution comprising high-performing cells. To that end, we first condition our model on the accuracy metric.

### 4.1.2 Conditional model

To achieve the generation of high-performing architectures, we propose a guidance approach, inspired by the classifier-free guidance (Ho & Salimans, 2021), and integrate it to our unconditional graph diffusion model.

Unlike the unconditional model, our conditional model estimates the distribution $p_\theta(\mathbf{G}^{t-1}|\mathbf{G}^t, y)$, essentially by computing the score function $\nabla_{\mathbf{G}^{t-1}} \log p_{\theta_\gamma}(\mathbf{G}^{t-1}|\mathbf{G}^t, y)$ as:

$$\nabla_{\mathbf{G}^{t-1}} \log p_{\theta_\gamma}(\mathbf{G}^{t-1}|\mathbf{G}^t, y) = (1-\gamma)\nabla_{\mathbf{G}^{t-1}} \log p_\theta(\mathbf{G}^{t-1}|\mathbf{G}^t) + \gamma\nabla_{\mathbf{G}^{t-1}} \log p_\theta(\mathbf{G}^{t-1}|\mathbf{G}^t, y), \qquad (9)$$

where $\gamma$ is the guidance scale and y is the target variable. The first term of Eq. 9 corresponds to the unconditional distribution $p_\theta(\mathbf{G}^{t-1}|\mathbf{G}^t)$ learning, while the second one corresponds to the conditional distribution $p_\theta(\mathbf{G}^{t-1}|\mathbf{G}^t, y)$ learning. Following Ho & Salimans (2021), we remove the conditioning information for some forward passes determined by the control parameter $\epsilon$. This leads to the unconditional training of the network. For the rest forward passes, we keep this information to enable conditional training.

### 4.1.3 Discretisation of the target variable

Our guidance approach assumes that the target variable $y$, e.g. accuracy or latency, belongs to a finite set such that $y \in \{y_1, y_2, \ldots, y_w\} \subseteq \mathbb{R}$, where $w$ is the number of possible values of $y$. However, in our case, $y$ takes continuous values from the real number domain, $\mathbb{R}^+$. To address this issue, we split $y$ into $d$ discrete classes based on their value. The choice of the split affects the balance of the class data distribution. In our implementation (Sec. 5.1), we provide information on how we select the number of classes and splits according to the problem.

## 4.2 Incorporating Multiple conditions

Next, we introduce multiple conditions to the diffusion guidance to impose several constraints. Consider the unconditional noise model $q$ that corrupts the data progressively for $t$ time steps. Our objective is to estimate the reverse conditional diffusion process $\hat{q}(\mathbf{G}^{t-1}|\mathbf{G}^t, y_1, y_2, ..., y_k)$, given the $k$ independent conditions $y_1, y_2, ..., y_k$. Assuming $p_\theta(\mathbf{G}^{t-1}|\mathbf{G}^t, y_1, y_2, ..., y_k)$ approximates $\hat{q}(\mathbf{G}^{t-1}|\mathbf{G}^t, y_1, y_2, ..., y_k)$, we perform the estimation of reverse conditional diffusion process by computing the score function $\nabla_{\mathbf{G}^{t-1}} \log p_{\theta_\gamma}(\mathbf{G}^{t-1}|\mathbf{G}^t, y_1, ..., y_k)$ as:

$$\begin{aligned} \nabla_{\mathbf{G}^{t-1}} \log p_{\theta_\gamma}(\mathbf{G}^{t-1}|\mathbf{G}^t, y_1, ..., y_k) = &(1-\gamma)\nabla_{\mathbf{G}^{t-1}} \log p_\theta(\mathbf{G}^{t-1}|\mathbf{G}^t) \\ &+ \gamma\nabla_{\mathbf{G}^{t-1}} \log p_\theta(\mathbf{G}^{t-1}|\mathbf{G}^t, y_1, ..., y_k), \end{aligned} \qquad (10)$$

where $\gamma$ is the guidance scale. The derivation is provided in Appendix A.1. Similar to the standard single-conditioned guidance (Eq. 9), the conditional score function for multi-conditioned guidance can be expressed as a weighted sum of conditional and unconditional score function. These score functions can be computed using two forward passes of our network, the unconditional and conditional forward pass.

## 4.3 Training and Sampling

**Training procedure** Let $c_1, \ldots, c_k$ denote the metrics, which we want to constrain e.g. accuracy or hardware latency. The training procedure starts by randomly selecting the time-step $t$ from the range $(1, .., T)$. Subsequently, the performance metrics $P = (c_1, ..., c_k)$ undergo a substitution with a null token $\emptyset$ for a probability of $\epsilon$ instances. Then, marginal noise is introduced to both $\mathbf{X}$ and $\mathbf{E}$ for a duration of $t$ time-steps. Next, each of $c_1, .., c_k$ is individually processed through distinct embeddings, with the resultant embeddings being included to both $\mathbf{X}$ and $\mathbf{E}$. We then apply positional encoding to $\mathbf{X}$. Finally, the resultant graph is provided as input to our Graph Transformer network $\phi_\theta$. This network then generates the denoised graph $(\mathbf{X}, \mathbf{E})$ which is used to calculate the loss (Eq. 6). We present the training algorithm in Alg. 1.

**Sampling procedure** Let $(\hat{c}_1, .., \hat{c}_k)$ be the constraints desired to be imposed (e.g. $\hat{c}_1$=top 5%). The sampling procedure is initiated with sampling a random noisy graph $\mathbf{G^t}$ from the prior distribution $(q_X(n_T) \times q_E(n_T))$. Next, we apply the positional encoding to X and perform two forward passes of our trained network $\phi_\theta$, namely the unconditional and conditional pass. In the unconditional pass, $(c_1 = \emptyset, c_2 = \emptyset, .., c_k = \emptyset)$, where $\emptyset$ is a null token, whereas for the conditional pass, $(c_1 = \hat{c}_1, ..., c_k = \hat{c}_k)$. Then, the score estimates are computed for both functions ($\hat{p}_c$ for conditional and $\hat{p}_u$ for unconditional). Lastly, we calculate the resulting

score by a linear combination of the score estimates and sample a less noisy graph $\mathbf{G}^{t-1}$ with Eq. 7 and 8. This is iteratively performed to produce the clean graph $\mathbf{G}^0$. The sampling algorithm is presented in Alg. 2 and implementation details in Appendix Sec. A.2.

---

**Algorithm 1** Training DiNAS

---

**Input**: $\mathbf{G}^0 = (\mathbf{X}, \mathbf{E}, c_1, ..., c_k)$ and $\epsilon$
$t \sim \nu(1, .., T)$          ▷ Sample t randomly from $(1, ...T)$
$c_1, ..., c_k \leftarrow \emptyset$ with probability $\epsilon$      ▷ Conditional dropout to train unconditionally
$\mathbf{G}^t \leftarrow (\mathbf{X}Q_X^t, \mathbf{E}Q_E^t, c_1, ..., c_k)$      ▷ Apply marginal noise for $t$ time steps
$\mathbf{G}^t \leftarrow (\mathbf{G}^t, Emb(c_1), ..., Emb(c_k))$      ▷ Append embeddings to nodes and edges
$\mathbf{X}^t \leftarrow \mathbf{X}^t + PosEnc(\mathbf{X}^t)$      ▷ Add sinusoids to X for positional encoding
$\hat{p}^X, \hat{p}^E \leftarrow p_\theta(\mathbf{G}^t | c_1 ..., c_k)$      ▷ Forward pass
$optimiser.step(L_G(\hat{p}^X, \mathbf{X}, \hat{p}^E, \mathbf{E}))$      ▷ Calculate loss and optimise (Eq. 6)

---

---

**Algorithm 2** Sampling from DiNAS

---

**Input**: guidance scale $\gamma$, and conditions $\hat{c}_1, .., \hat{c}_k$
Sample $n_T$ number of nodes from training data distribution
Sample random graph $\mathbf{G}^t \sim (q_X(n_T) \times q_E(n_T))$      ▷ Sample from prior distribution
**for** $t = T$ to 1 **do**
    $\mathbf{X}^t \leftarrow \mathbf{X}^t + PosEnc(\mathbf{X}^t)$      ▷ Add sinusoids to X for positional encoding
    $\hat{p}_u^X, \hat{p}_u^E = p_\theta(\mathbf{G}^t | c_1 = \emptyset, ..., c_k = \emptyset)$      ▷ Unconditional forward pass
    $\hat{p}_c^X, \hat{p}_c^E = p_\theta(\mathbf{G}^t | c_1 = \hat{c}_1, .., c_k = \hat{c}_k)$      ▷ Conditional forward pass
    $\hat{p}^X = (1 - \gamma)\hat{p}_u^X + \gamma\hat{p}_c^X$      ▷ Linear combination of score estimates
    $\hat{p}^E = (1 - \gamma)\hat{p}_u^E + \gamma\hat{p}_c^E$      ▷ Linear combination of score estimates
    Calculate $p_\theta(x_i^{t-1} | \mathbf{G}^t)$ and $p_\theta(e_{ij}^{t-1} | \mathbf{G}^t)$      ▷ Eq. (7)
    $\mathbf{G}^{t-1} \sim \prod_i p_\theta(x_i^{t-1} | \mathbf{G}^t) \prod_{ij} p_\theta(e_{ij}^{t-1} | \mathbf{G}^t)$      ▷ Sample $\mathbf{G}^{t-1}$ (Eq. 8)
**end for**
**return** $\mathbf{G}^0$

---

## 5 Experiments

We evaluate our approach on six standard benchmarks- encompassing tabular, surrogate, hardware aware benchmarks, and the challenging ImageNet image classification task (Deng et al., 2009).

### 5.1 Experimental Setup

**Tabular Benchmarks** We first consider the tabular benchmarks- NAS-Bench-101 (Ying et al., 2019) and NAS-Bench-201 (Dong & Yang, 2020) for our experiments. Tabular benchmarks list unique architectures with their corresponding accuracy. We utilise the validation accuracy as performance metrics $P$. The evaluation protocol [2] follows the established standard (Yan et al., 2020; Wu et al., 2021) of conducting a search for the maximum validation accuracy within a fixed number of queries and reporting the corresponding test accuracy, both as a mean over 10 runs. Although there are different other factors (like the nature of the algorithm) affecting the search time for different approaches, generally search times are directly proportional to the number of queries, and is thus used as an efficiency metric by previous approaches by Lukasik et al. (2022), Yan et al. (2020) and Wu et al. (2021).

For NAS-Bench-101, we compare our approach with Arch2Vec (Yan et al., 2020), NAO (Luo et al., 2018), BANANAS (White et al., 2021a), Bayesian Optimisation (Snoek et al., 2015), Local Search (White et al., 2021b), Random Search (Li & Talwalkar, 2020), Regularised Evolution (Real et al., 2019), WeakNAS (Wu

---

[2]The detailed evaluation protocol for each benchmark can be found in Appendix A.5.

et al., 2021) and AG-Net (Lukasik et al., 2022). For NAS-Bench-201, our approach is evaluated against SGNAS (Huang & Chu, 2021), GANAS (Rezaei et al., 2021), BANANAS, Bayesian Optimisation, Random Search, AG-Net, TNAS (Shala et al., 2023), MetaD2A (Lee et al., 2021), $\beta$-DARTS (Ye et al., 2022) and DiffusionNAG (An et al., 2023). The corresponding results are reported in Tables 1 and 2.

**Surrogate Benchmarks** Next, we evaluate our method on surrogate benchmarks. Surrogate benchmarks operate on significantly larger search spaces like DARTS (Liu et al., 2019) or NAS-Bench-NLP (Klyuchnikov et al., 2020) and therefore use a simple surrogate predictor to estimate the ground truth accuracy. We perform our experiments on two surrogate benchmarks, the NAS-Bench-301 (Siems et al., 2021) (trained on CIFAR-10 (Krizhevsky et al., 2009)) on DARTS search space and NAS-Bench-NLP. We report the maximum validation accuracy as a mean over 10 runs, along with the number of queries and compare our method to the prior work as with NAS-Bench-101. The results are presented in Table 3.

**Hardware Aware Benchmark** Our next evaluation is on the Hardware Aware Benchmark (HW-NAS-Bench) (Li et al., 2021). HW-NAS-Bench provides hardware information (e.g. latency) along with the accuracy

Table 1: Comparison of results on NAS-Bench-101. 'Val' represents the maximum validation accuracy and 'Test' represents the corresponding test accuracy, both as a mean over 10 runs. Queries are the number of retrieval attempts for accuracy from the benchmark.

| Methods | Val(%) | Test(%) | Queries ↓ |
|---|---|---|---|
| Optimum | 95.06 | 94.32 | |
| Arch2vec + RL | - | 94.10 | 400 |
| Arch2vec + BO | - | 94.05 | 400 |
| NAO [†] | 94.66 | 93.49 | 192 |
| BANANAS [†] | 94.73 | 94.09 | 192 |
| Local Search [†] | 94.57 | 93.97 | 192 |
| Random Search [†] | 94.31 | 93.61 | 192 |
| Bayesian Optimisation [†] | 94.57 | 93.96 | 192 |
| WeakNAS | - | 94.18 | 200 |
| Regularised Evolution [†] | 94.47 | 93.89 | 192 |
| AG-Net | 94.90 | 94.18 | 192 |
| **DiNAS (ours)** | **94.98** | **94.27** | **150** |

for multiple edge devices. We follow the standard protocol (Lukasik et al., 2022) and report the accuracy of best found architectures for ImageNet classification task given the latency constraint (in milliseconds) as a mean over 10 runs along with the number of queries. We also report the feasibility, which indicates the percentage of generated architectures following the given latency constraint. We compare our approach to Random Search and AG-Net as strong baselines for multiple devices each in multiple latency constraints. The results are available in Table 11.

**Experiments on ImageNet** Lastly, we conduct experiments on the large-scale image classification task ImageNet (Deng et al., 2009), following the protocol from Liu et al. (2019); Chen et al. (2021a). This involves training and evaluating the best generated architecture from NASBench301 (trained on CIFAR10 image classification task) on the ImageNet dataset. We report the top-1 and top-5 errors along with the number of queries, comparing our method to several robust baselines (e.g. DARTS, TENAS (Chen et al., 2021a), NASNET-A (Zoph et al., 2018) , DrNAS (Chen et al., 2021b), $\beta$-DARTS (Ye et al., 2022), Sweetimator (Yang et al., 2023), DARTS $^{++}$ (Soro & Song, 2023), and AG-Net. To ensure a fair comparison, we report the results of methods with search on CIFAR-10 and evaluation on ImageNet, wherever applicable. We summarise the results in Table 5.

**Implementation** The discretization of the target variable is essential for our task as it is not sufficient to train our model solely on high-performing samples due to the low number of high-performing architectures. We empirically found that in our task, $d = 2$ for the accuracy metric has a slightly superior performance over other values of $d$ (see ablation study in Appendix Sec. 5.3.4) and thus, we discretise the accuracy into two classes. One class includes $> f_{th}$ percentile of accuracy values, while the remaining values belong to the other class. Using higher values of $f$ for accuracy generates better-performing architectures, but they also lead to class imbalance, thereby reducing the model performance. We address this issue by modifying $f$ depending on the data availability of the specific benchmark. For generating high performing samples, the model is conditioned to generate the samples belonging to $> f_{th}$ percentile class for accuracy during

Table 2: Comparison of results on NAS-Bench-201 for different datasets. 'Val' represents the maximum validation accuracy and 'Test' represents the corresponding test accuracy, both as a mean over 10 runs. Queries/Gen. are the number of retrieval attempts for accuracy from the benchmark or number of architecture generations

| Methods | CIFAR-10 | | CIFAR-100 | | ImageNet16-120 | | Queries/Gen. ↓ |
|---|---|---|---|---|---|---|---|
| | Val(%) | Test (%) | Val(%) | Test(%) | Val(%) | Test(%) | |
| **Optimum\*** | 91.61 | 94.37 | 73.49 | 73.51 | 46.77 | 47.31 | - |
| SGNAS | 90.18 | 93.53 | 70.28 | 70.31 | 44.65 | 44.98 | - |
| BANANAS [†] | 91.56 | 94.30 | 73.49* | 73.50 | 46.65 | 46.51 | 192 |
| Bayesian Opt.[†] | 91.54 | 94.22 | 73.26 | 73.22 | 46.43 | 46.40 | 192 |
| Random Search[†] | 91.12 | 93.89 | 72.08 | 72.07 | 45.97 | 45.98 | 192 |
| GANAS | - | 94.34 | - | 73.28 | - | **46.80** | 444 |
| AG-Net | 91.60 | 94.37* | 73.49* | 73.51* | 46.64 | 46.43 | 192 |
| TNAS | - | 94.37* | - | 73.51* | - | - | - |
| MetaD2A | - | 94.37* | - | 73.34 | - | - | 500 |
| $\beta$-DARTS | 91.55 | 94.36 | 73.49* | 73.51* | 46.37 | 46.34 | - |
| DiffusionNAG | - | 94.37* | - | 73.51 | - | - | |
| **DiNAS (ours)** | **91.61\*** | **94.37\*** | **73.49\*** | **73.51\*** | **46.66** | 45.41 | 192 |

Table 3: Comparison of results on NAS-Bench-301 (left) and NAS-Bench-NLP (right). 'Val' represents the maximum validation accuracy as a mean over 10 runs. Queries are the number of retrieval attempts for accuracy from the benchmark.

| Methods | NAS-Bench-301 | | NAS-Bench-NLP | |
|---|---|---|---|---|
| | Val(%) | Queries↓ | Val(%) | Queries↓ |
| BANANAS[†] | 94.47 | 192 | 95.68 | 304 |
| Bayesian Opt.[†] | 94.71 | 192 | - | - |
| Random Search[†] | 94.31 | 192 | 95.64 | 304 |
| Regularised Evolution[†] | 94.75 | 192 | 95.66 | 304 |
| AG-Net [‡] | 94.79 | 192 | 95.95 | 304 |
| **DiNAS (ours)** | **94.92** | **100** | **96.06** | 304 |

the sampling process. Specific values for each benchmark can be found in the Appendix Sec. A.5. For the metric of latency in the hardware-aware benchmark (Li et al., 2021), we wish to generate high-performing architectures lower than the given latency constraint. To achieve this, we discretize latency into two discrete classes- one below the constraint value and one above the constraint.

## 5.2 Discussion of Results

**Tabular Benchmarks** Tables 1 and 2 present empirical evidence of the superior performance of our proposed method in the context of tabular benchmarks. Across both tabular benchmarks, our approach consistently outperforms the SOTA or converges to optimal validation accuracy. Notably, for NAS-Bench-101, our method concurrently reduces query count by 25%, demonstrating its effectiveness. GANAS exhibits a slightly better test accuracy on ImageNet in NAS-Bench-201 experiments, which can be explained by the fact that our method searches for the best architecture in terms of validation accuracy and the best validation accuracy does not necessarily imply best test accuracy. To that end, we found that some of the generated architectures with high validation accuracy correspond to a comparatively low test accuracy for the case of ImageNet. This discrepancy is also reflected in the standard deviation values for ImageNet, reported in Table 10.

**Surrogate Benchmarks** Furthermore, the results in Table 3 demonstrate that DiNAS excels in surrogate benchmarks as well. Our method achieves the SOTA in nearly 50% reduction in queries for NAS-Bench-301 and the same query count in NAS-Bench-NLP, surpassing the performance of previous methods such as Random Search, Bayesian Optimisation, and AG-Net. The results from NAS-Bench-NLP experiment also

Table 4: Comparison of results on HW-NAS-Bench. 'Val' represents the maximum validation accuracy as a mean over 10 runs and 'Feas' represents the feasibility considering generations of all the runs. Queries are the number of retrieval attempts for accuracy and latency from the benchmark.

| Device | Lat. (ms) | DiNAS | | AG-Net | | Random | Queries ↓ |
|---|---|---|---|---|---|---|---|
| | | Val(%) | Feas. (%)↑ | Val(%) | Feas. (%)↑ | Val(%) | |
| EdgeGPU | 2 | 39.44 | **92.60** | 39.70 | 29.00 | 37.20 | 200 |
| | 4 | **43.91** | **93.20** | 42.80 | 29.00 | 41.70 | 200 |
| | 6 | 45.03 | **66.35** | 45.30 | 64.00 | 44.90 | 200 |
| Raspi4 | 2 | **34.67** | **92.80** | 34.60 | 28.00 | 33.90 | 200 |
| | 4 | **43.25** | **77.80** | 42.00 | 47.00 | 41.90 | 200 |
| | 6 | **44.72** | **57.70** | 44.00 | 56.00 | 43.20 | 200 |
| EdgeTPU | 1 | 45.31 | 48.37 | 46.40 | 74.00 | 45.40 | 200 |
| Pixel3 | 2 | 40.01 | **97.30** | 40.90 | 48.00 | 38.80 | 200 |
| | 4 | 44.74 | **82.50** | 45.30 | 69.00 | 43.8 | 200 |
| | 6 | **45.95** | **78.50** | 45.70 | 77.00 | 45.1 | 200 |
| Eyeris | 1 | **44.67** | **78.12** | 44.50 | 49.00 | 43.30 | 200 |
| FPGA | 1 | **44.53** | **91.65** | 43.30 | 65.00 | 42.90 | 200 |

prove that our approach is not only effective in image classification tasks but also in NLP tasks, proving our approach to be task-independent.

**Hardware-Aware benchmark** From Table 11, we can observe that our approach outperforms Random Search in most cases and surpassing AG-Net in over half of them. Additionally, our method excels in feasibility across diverse devices and latency constraints while using the same number of queries compared to AG-Net, proving that our multi-conditioned guidance was indeed able to replicate the behaviour of multiple independent predictors (for accuracy and latency) using a single set of hyperparameters.

**ImageNet** Lastly, we can observe from Table 5 that our approach demonstrates competitive performance with low top-1 and top-5 error rates on ImageNet, outperforming robust baselines such as DARTS, NASNET-A and TENAS. However, the generations from AG-Net , DrNAS (Chen et al., 2021b), DARTS [++] Soro & Song (2023), $\beta$-DARTS (Ye et al., 2022), and Sweetimator (Yang et al., 2023) are better performing on ImageNet than our method, with the best-performing methods being DrNAS and $\beta$-DARTS. The comparison of our approach to non-generative methods gives our approach a disadvantage due to the unavailability of a dataset with DARTS-style normal-reduced cell architectures and ImageNet accuracies. Although the transferability works well in this case, it does not beat the state-of-the-art unfortunately. This experiment thus highlights that the generated architectures from our method possess generalisation capabilities across different datasets.

## 5.3 Ablation studies

### 5.3.1 Comparison of search times

To prove the efficiency of our method, we report the training (one-time cost) and search/generation times for the search on CIFAR-100 and ImageNet datasets in the Table 6. The search times correspond to generating all the architectures in one single run of an experiment. We observe that our approach requires a significantly less search time compared to previous approaches on ImageNet. On CIFAR100 dataset, our approach outperforms DiffusionNAG An et al. (2023) by a significant margin. It can thus be proven that our approach, without an external classifier, is more efficient than predictor-based approaches, due to its reduced computational requirement for each generation.

The search time comparisons are concurrent with the protocol from other generative NAS methods by Lukasik et al. (2022) and Huang & Chu (2021), where the search times are taken as the generation times of

[‡]Note that Lukasik et al. (2022) refers to validation accuracy of NAS-Bench-NLP as validation perplexity
[†]Results taken from Lukasik et al. (2022)

Table 5: Comparison of results for top-1, top-5 errors on ImageNet.

| Methods | Top-1 ↓ | Top-5 ↓ | Queries↓ |
|---|---|---|---|
| NASNET-A | 26.0 | 8.4 | 20000 |
| DARTS | 26.7 | 8.7 | - |
| DrNAS | **23.7** | 7.1 | - |
| TENAS | 26.2 | 8.3 | - |
| AG-Net | 24.1 | 7.2 | 304 |
| Sweetimator | 24.1 | - | - |
| DARTS$^{++}_{aug}$ | 24.8 | 7.8 | - |
| $\beta$-DARTS | 23.9 | **7.0** | - |
| **DiNAS (ours)** | 24.8 | 7.4 | **100** |

Table 6: Comparison of search times in GPU seconds and training times of the pre-trained generator (if applicable) in GPU hours on ImageNet and CIFAR 100 (using NAS-Bench-201) for different approaches. The search times correspond to generating all the architectures in one single run of an experiment. Note that we report the mean time over different runs for DiNAS.

| Dataset | Methods | Search/Gen. Time (GPU sec.)↓ | Pre-training cost (GPU hrs) ↓ |
|---|---|---|---|
| ImageNet | NASNET-A | $1.7 \times 10^8$ | - |
| ImageNet | DARTS | 345600 | - |
| ImageNet | DrNAS | 397440 | - |
| ImageNet | TENAS | 4320 | - |
| ImageNet | AG-Net | 1728 | 21.6 |
| ImageNet | $\beta$-DARTS | 34560 | - |
| ImageNet | DARTS$^{++}_{aug}$ | 25920 | - |
| ImageNet | **DiNAS (ours)** | **53.76** | **16.6** |
| CIFAR100 | DiffusionNAG | 261 | 3.43 |
| CIFAR100 | **DiNAS (ours)** | **15.36** | **0.25** |

the proposed method. The training times in Table 6 are considered as a one-time cost because the search time of a pre-trained model is the inference cost of the model. Upon training, the pre-trained model can then be used to generate architectures, which also generalise to different datasets (see Table 5).

### 5.3.2 Novelty and Uniqueness Analysis

We conduct an analysis of novelty and uniqueness for the generated architectures. We start by generating 2000 and 100,000 architectures respectively based on our proposed method. To assess novelty, we calculate the percentage of generated samples absent in the training data whereas to assess uniqueness, we calculate the ratio of architectures present just once in the generations to the total number of generations(Zhang et al., 2019) (An et al., 2023). Given the enormous size of DARTS and NAS-Bench-NLP search spaces, we consider NAS-Bench-301 and NAS-Bench-NLP for our analysis. Furthermore, to examine the efficiency of our method, we record and report the training times for our method (for 100 epochs), along with the sampling times per architecture using a single NVIDIA A6000 GPU on five different benchmarks. We can observe the results of our ablation studies in Table 7. Note that for benchmarks involving multiple cases (e.g. HWNAS and NAS-Bench-201), we take the mean of the training times for all cases.

We observe from the results of the novelty analysis for the case of 2000 samples and 100,000 samples that in both the datasets, all the generated samples are novel and most of them are unique, proving that our method is not just selecting the best-performing architectures from the training set. We can also observe that while the novelty for all the cases remain at the top, the uniqueness suffers a bit when sampling high number of architectures, due to a limited number of possible high-performing architectures. Moreover, we can observe that the sampling rates of each benchmark are in milliseconds, proving the rapid generation capabilities of our method.

Table 7: Ablation study on novelty analysis (left) and efficiency analysis (right). Note that 'Nov' represents the novelty ratio, 'Uni' represents the uniqueness ratio and 'Gen' represents the number of architectures generated. The training time is reported in hours and sampling time per architecture in seconds.

| Benchmark | Gen. | Nov.(%) ↑ | Uni.(%) ↑ |
|-----------|------|-----------|-----------|
| NAS-Bench-301 | 2000 | 100 | 97.37 |
| | 100,000 | 100 | 91.10 |
| NAS-Bench-NLP | 2000 | 100 | 97.57 |
| | 100,000 | 100 | 96.78 |

| Benchmark | Train (hrs)↓ | Sample (sec)↓ |
|-----------|--------------|---------------|
| NB101 | 0.96 | 0.09 |
| NB201 | 0.25 | 0.08 |
| NB301 (Normal) | 8.3 | 0.14 |
| NB301 (Reduced) | 8.3 | 0.14 |
| NBNLP | 0.95 | 0.15 |
| HWNAS | 1.5 | 0.08 |

### 5.3.3 Required number of training samples

This section analyses the performance of our method on DARTS search space (Liu et al., 2019) when trained on different number of samples. In particular, we consider training in three scenarios- on 100,000 samples, 10,000 samples and 1,000 samples, for which, we randomly sample the given number of training samples from DARTS search space and follow the same training protocol as our experiments on NAS-Bench-301. Upon training in each case, 192 architectures are generated for a total of 10 runs and the maximum validation accuracy is reported as a mean over these runs. In addition, we calculate and report the novelty and uniqueness of the generated architectures, calculated using the same methodology as Section 5.3.2 considering generations from all the 10 runs. Finally, we report the number of queries. The results for this ablation study are reported in Table 8.

Table 8: Peformance on NAS-Bench-301 (Siems et al., 2021) with different number of samples. The Val. acc represents the maximum validation accuracy, reported as a mean over 10 runs, whereas novelty and uniqueness are calculated considering the generations from all the runs. Queries are the number of retrieval attempts for accuracy from the benchmark.

| Training Samples | Val acc.(%) ↑ | Novelty(%) ↑ | Uniqueness(%) ↑ | Queries |
|------------------|---------------|--------------|-----------------|---------|
| 100,000 | 94.92 | 100 | 97.37 | 192 |
| 10,000 | 94.89 | 100 | 100 | 192 |
| 1,000 | 85.29 | 100 | 100 | 192 |

We observe from Table 8 that our method learns the architectural representation and finds the high-performing architectures with one tenth of the number of training samples as well. However, the mean maximum validation accuracy drops when we further reduce the training samples to 1000. We found out that the reason for this decline was the unavailability of any valid generations in one of the runs. This resulted in the maximum validation accuracy for that run to be 0, which influenced the mean. From this, we can conclude that the data capture capabilities of our model are compromised when training on a very small number of samples. Moreover, we observe that the novelty and uniqueness ratios do not suffer at all when reducing the training data availability.

### 5.3.4 Number of classes for guidance

The goal of this ablation study is to analyse the effect of the number of classes $d$ for accuracy present in the training data on our approach. We train and evaluate our method on NAS-Bench-201 (Dong & Yang, 2020) for the task of CIFAR-10 image classification involving four different cases: $d = \{2, 3, 4, 5\}$. We use the same training and evaluation protocol as experiments in Table 2. The split of the data into classes, denoted as $s_T$, is performed depending on specific percentiles $f = (f_1, f_2, \ldots, f_{d-1})$ of the accuracy. For instance, for two classes, $f = 95$ and the split $s_T = [95_{th} - 100_{th}, 0_{th} - 95_{th}]$, while for three classes, $f = [80, 95]$ and $s_T = [95_{th} - 100_{th}, 80_{th} - 95_{th}, 0_{th} - 80_{th}]$. In all the cases, we generate architectures belonging to the class of >95th percentile. The choice of $f$ is empirical as it does not affect the samples in the class we want to condition our model on (i.e. $> 95_{th}$ percentile). We report the maximum validation accuracy and the

corresponding test accuracy, both as a mean over 10 runs, along with the number of queries. Moreover, we report the percentiles $f$ used for splitting. The results for this study are reported in Table 9.

Table 9: Comparison of results on NAS-Bench-201 for CIFAR-10 when using different number of classes. Here, $f$ represents the percentiles used for splitting the data into classes, 'Val' and 'Test' represent the maximum validation accuracy and the corresponding test accuracy, both represented as a mean over 10 runs. Queries are the number of retrieval attempts for accuracy from the benchmark.

| Number of classes ($d$) | $f$ | Val(%)↑ | Test(%)↑ | Queries |
|---|---|---|---|---|
| 2 | [95] | **91.61** | **94.37** | 192 |
| 3 | [80, 95] | 91.60 | 94.00 | 192 |
| 4 | [50, 80, 95] | 91.52 | 93.79 | 192 |
| 5 | [30, 50, 80, 95] | 91.57 | 93.89 | 192 |

We observe from Table 9 that discretising the accuracy into two classes results in a slightly superior performance over other values of $d$. However, the differences are marginal.

## 6  Conclusion

We presented a generative method to facilitate the search process for neural architectures. Our approach uses a conditional graph diffusion model to rapidly generate novel, unique and high-performing neural network architectures. In this context, we first formulated the classifier-free guidance for graph diffusion models and then proposed a multi-conditioned classifier-free guidance for diffusion models. Unlike the related work, our method does not require an external surrogate predictor and is thus differentiable. In the experiments, we demonstrated state-of-the-art performance in tabular, surrogate and hardware-aware evaluations by considering six standard benchmarks. Furthermore, we have shown the search efficiency of our approach compared to the previous work using ablation studies. We observed that our method samples architectures two orders of magnitude faster than other generative NAS approaches and at least three orders of magnitude faster than classic approaches.

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

# A Appendix

## A.1 Proofs and Derivations

**Derivation 1:** Let $q$ be the unconditional Markovian noise model and $\hat{q}$ be the conditional noising process similar to $q$. We define our aim as decomposing $\hat{q}(\mathbf{G}^{t-1}|\mathbf{G}^t, y_1, y_2, ..., y_k)$ and then deriving the score function

for the multi-conditioned diffusion process. We start by expanding the term:

$$\hat{q}(\mathbf{G}^{t-1}|\mathbf{G}^t, y_1, y_2, \ldots, y_k) = \frac{\hat{q}(\mathbf{G}^{t-1}, \mathbf{G}^t, y_1, y_2, \ldots, y_k)}{\hat{q}(\mathbf{G}^t, y_1, y_2, \ldots, y_k)} \tag{11}$$

$$= \frac{\hat{q}(\mathbf{G}^{t-1}, \mathbf{G}^t, y_1, y_2, \ldots, y_k)}{\hat{q}(y_1, y_2, \ldots, y_k|\mathbf{G^t})\hat{q}(\mathbf{G^t})} \tag{12}$$

$$= \frac{\hat{q}(y_1, \ldots, y_k|\mathbf{G}^{t-1}, \mathbf{G}^t)\hat{q}(\mathbf{G}^{t-1}, \mathbf{G}^t)}{\hat{q}(y_1, y_2, \ldots, y_k|\mathbf{G^t})\hat{q}(\mathbf{G^t})} \tag{13}$$

$$= \frac{\hat{q}(y_1, \ldots, y_k|\mathbf{G}^{t-1}, \mathbf{G}^t)\hat{q}(\mathbf{G}^{t-1}|\mathbf{G}^t)\hat{q}(\mathbf{G}^t)}{\hat{q}(y_1, y_2, \ldots, y_k|\mathbf{G^t})\hat{q}(\mathbf{G^t})} \tag{14}$$

$$= \frac{\hat{q}(y_1, \ldots, y_k|\mathbf{G}^{t-1}, \mathbf{G}^t)\hat{q}(\mathbf{G}^{t-1}|\mathbf{G}^t)}{\hat{q}(y_1, y_2, \ldots, y_k|\mathbf{G^t})} \tag{15}$$

Dhariwal & Nichol (2021) prove that the classification term (in our case $\hat{q}(y_1, \ldots, y_k|\mathbf{G}^{t-1}, \mathbf{G}^t)$) does not depend on the noisier version of G (i.e. $\mathbf{G^t}$) and can be rewritten as $\hat{q}(y_1, \ldots, y_k|\mathbf{G}^{t-1})$. Furthermore, they also show that $\hat{q}$ behaves the same as $q$ when not conditioned on the classification targets $y_1, \ldots, y_k$. We use these findings to further simplify Eq. 15 to:

$$\hat{q}(\mathbf{G}^{t-1}|\mathbf{G}^t, y_1, y_2, \ldots, y_k) = \frac{\hat{q}(y_1, \ldots, y_k|\mathbf{G}^{t-1})q(\mathbf{G}^{t-1}|\mathbf{G}^t)}{\hat{q}(y_1, y_2, \ldots, y_k|\mathbf{G^t})} \tag{16}$$

We assume that the generative model $p_\theta(\mathbf{G}^{t-1}|\mathbf{G}^t)$ approximates $q(\mathbf{G}^{t-1}|\mathbf{G}^t)$ and the classifier $p_\psi(y_1, \ldots, y_k|\mathbf{G}^{t-1})$ approximates $\hat{q}(y_1, \ldots, y_k|\mathbf{G}^{t-1})$. By substituting the distributions with their approximations, taking the logarithm and calculating the gradients w.r.t. $\mathbf{G}^{t-1}$, we obtain the following score function:

$$\nabla_{\mathbf{G}^{t-1}} \log p_{\theta, \psi}(\mathbf{G}^{t-1}|\mathbf{G}^t, y_1, \ldots, y_k) = \nabla_{\mathbf{G}^{t-1}} \log p_\theta(\mathbf{G}^{t-1}|\mathbf{G}^t) + \nabla_{\mathbf{G}^{t-1}} \log p_\psi(y_1, \ldots, y_k|\mathbf{G}^{t-1}), \tag{17}$$

where $\mathbf{G^t}$ and $\mathbf{G}^{t-1}$ represent the DAG $\mathbf{G}$ at time step $t$ and $t-1$ respectively, $\psi$ are the classifier parameters and $\theta$ are the generative model parameters. Similar to the standard classifier-based guidance (Dhariwal & Nichol, 2021), we multiply the conditioning term by a factor of $\gamma$. Thus, we can express the reverse denoising process as a weighted sum of the unconditional score function $\nabla_{\mathbf{G}^{t-1}} \log p_\theta(\mathbf{G}^{t-1}|\mathbf{G}^t)$ and the conditioning term $\nabla_{\mathbf{G}^{t-1}} \log p_\psi(y_1, \ldots, y_k|\mathbf{G}^{t-1})$, given by:

$$\nabla_{\mathbf{G}^{t-1}} \log p_{\theta_\gamma, \psi}(\mathbf{G}^{t-1}|\mathbf{G}^t, y_1, \ldots, y_k) = \nabla_{\mathbf{G}^{t-1}} \log p_\theta(\mathbf{G}^{t-1}|\mathbf{G}^t) + \gamma \nabla_{\mathbf{G}^{t-1}} \log p_\psi(y_1, \ldots, y_k|\mathbf{G}^{t-1}), \tag{18}$$

where $\gamma$ is the guidance scale. Then, by substituting the conditioning term $\nabla_{\mathbf{G}^{t-1}} \log p_\psi(y_1, \ldots, y_k|\mathbf{G}^{t-1})$ from Eq. 17 and removing the classifier parameters $\psi$, we obtain:

$$\begin{aligned} \nabla_{\mathbf{G}^{t-1}} \log p_{\theta_\gamma}(\mathbf{G}^{t-1}|\mathbf{G}^t, y_1, \ldots, y_k) = {} & (1 - \gamma)\nabla_{\mathbf{G}^{t-1}} \log p_\theta(\mathbf{G}^{t-1}|\mathbf{G}^t) \\ & + \gamma \nabla_{\mathbf{G}^{t-1}} \log p_\theta(\mathbf{G}^{t-1}|\mathbf{G}^t, y_1, \ldots, y_k). \end{aligned} \tag{19}$$

Hence, as we can observe from Equation 19, we have successfully derived the score function of multi-conditioned diffusion guidance.

## A.2 Implementation Details

Our proposed method involves training a Graph Transformer network, proposed by Dwivedi & Bresson (2021), for denoising. This network comprises of an input node/edge wise MLP layer, followed by 5 Graph Transformer layers and, node/edge wise MLP as the output layer. Each Graph Transformer layer consists

Table 10: Mean validation and test accuracies with standard deviation for all runs of NAS-Bench-101, NAS-Bench-201, NAS-Bench-301 and NAS-Bench-NLP experiments. StD represents the standard deviation in the table.

| Experiment | Dataset | Val. acc. | | Test acc. | |
|---|---|---|---|---|---|
| | | Mean | StD | Mean | StD |
| NAS-Bench-101 | CIFAR-10 | 94.98 | 0.169 | 94.27 | 0.197 |
| NAS-Bench-201 | CIFAR-10 | 91.61 | 0.000 | 94.37 | 0.184 |
| NAS-Bench-201 | CIFAR-100 | 73.49 | 0.000 | 73.51 | 0.000 |
| NAS-Bench-201 | ImageNet | 46.66 | 0.092 | 45.41 | 0.589 |
| NAS-Bench-301 | CIFAR-10 | 94.92 | 0.072 | - | - |
| NAS-Bench-NLP | Penn Tree Bank | 96.06 | 0.173 | - | - |

of three main parts: a self-attention module similar to the one found in the standard Transformer model (Vaswani et al., 2017a), a fully connected layer, and layer normalisation. All the models were trained for 100 epochs with a learning rate of 0.0002, batch-size of 16, weight decay of $10^{-12}$, guidance scale of -4 and using AdamW optimiser (Loshchilov & Hutter, 2017) on a single NVIDIA A6000 GPU. For noising, we use cosine noise schedule for $T = 500$ time-steps. The hyperparameters used in our method were derived from the code from Vignac et al. (2023).

For the evaluation on ImageNet, we employ the same training pipeline and code as AG-Net (Lukasik et al., 2022) and TENAS (Chen et al., 2021a), taken from Chen (2022). We train the best generated architecture in terms of validation accuracy from NAS-Bench-301 on ImageNet for 250 epochs. The initial learning rate is set to 0.5 with a cosine learning rate scheduler and the batch size is set to 1024. The ImageNet training is performed on 3 NVIDIA V100 GPUs parallelly in a distributed manner.

## A.3 Additional Results

Table 10 provides some additional results, mainly reporting the mean and standard deviations over different runs for NAS-Bench-101, NAS-Bench-201, NAS-Bench-301 and NAS-Bench-NLP experiments. It can be observed that the standard deviation (StD.) for test accuracy is generally higher than the StD. for validation accuracy, particularly for NAS-Bench-201 on ImageNet. The reason for this difference is that the model is trained to generate architectures with high validation accuracy and has no test accuracy information. This also explains the discrepancy of the validation accuracy values and test accuracy values for the case of ImageNet in Table 2.

## A.4 Additional ablation studies

### A.4.1 Effect of the guidance scale parameter

We perform an additional ablation study to analyse the effect of the guidance scale parameter $\gamma$ on the performance of our model. We train our model on two differently sized search spaces: NAS-Bench-101 (size of 423k samples) and NAS-Bench-NLP (size of $10^{53}$ samples) search space using four guidance scales (-4, -2, 2 and 4) in this study and report the mean and standard deviation of the validation accuracy (and test accuracy for NAS-Bench-101) over 10 runs. The evaluation protocol is identical to the main experiments (Section 5).

Table 11 presents the results of this ablation study. Interestingly, we can observe that negative values of guidance scale perform better than positive values, unlike Ho & Salimans (2021). This can be attributed to the difference in formulation of Eq. 19 in Ho & Salimans (2021), where the guidance parameter is $w$, instead of $\gamma$ ($w \sim -\gamma$). We found that $\gamma = -4$ setting produces best results for both, small and large search spaces, and is thus used in our experiments.

Table 11: Comparison of performance of DiNAS using different guidance scales on NAS-Bench-101 and NAS-Bench-NLP search spaces. Val. represents the mean validation accuracy, 'Test' represents the mean test accuracy and StD. represents the standard deviation over 10 runs.

| $\gamma$ | NAS-Bench-101 | | NAS-Bench-NLP |
|---|---|---|---|
| | Val(%) $\pm$StD.$\uparrow$ | Test(%) $\pm$StD.$\uparrow$ | Val(%) $\pm$StD.$\uparrow$ |
| -4 | 94.98$\pm$0.17 | **94.27$\pm$0.20** | **96.06$\pm$0.17** |
| -2 | **95.06$\pm$0.14** | 94.16$\pm$0.20 | 95.97$\pm$0.12 |
| 2 | 92.60$\pm$0.43 | 92.05$\pm$0.58 | 95.42$\pm$0.08 |
| 4 | 91.44$\pm$0.47 | 90.86$\pm$0.59 | 95.36$\pm$0.29 |

## A.5 Evaluation protocols

**NAS-Bench 101 and NAS-Bench 201** We generate a fixed number of architectures (equal to the respective number of queries) and query them on both the benchmarks to find the maximum validation accuracy and its corresponding test accuracy. This process is repeated 10 times to calculate the mean maximum validation accuracy and mean corresponding test accuracy. Note that we use $f = 99$ for NAS-Bench-101 experiments and $f = 95$ for NAS-Bench-201 experiments.

**NAS-Bench-301** To evaluate our approach on NAS-Bench-301, a random subset of 100,000 architectures is selected from the DARTS search space. As surrogate benchmarks do not provide accuracy, the accuracy of the selected architectures are calculated using a pre-trained surrogate predictor XG-Boost (Chen & Guestrin, 2016) provided with NAS-Bench-301. Next, the network is trained using normal cells from this dataset, producing 10 normal cells from $> f_{th}$ class. Next, this process is repeated to produce 10 reduction cells. The evaluation involves 100 queries, considering all possible combinations of the 10 generated normal and 10 reduction cells. For each query, the highest validation accuracy and its corresponding test accuracy are recorded. This entire process is iterated 10 times, yielding mean values for these recorded accuracies. We use $f = 99$ for this benchmark.

**NAS-Bench-NLP** Given the enormity of this search space, we employ NAS-Bench-X11 (Yan et al., 2021b) as a surrogate predictor to obtain accuracy for these architectures, trained specifically on the Penn TreeBank dataset (Marcus et al., 1993). However, it should be noted that NAS-Bench-X11 is only capable of handling graphs with up to 12 nodes, which filters our dataset to include the total of 7,258 architectures. After training, we generate 304 architectures and estimate their accuracy using NAS-Bench-X11. The process is repeated 10 times. We set $f = 99$ for this benchmark.

**HW-NAS-Bench** For this evaluation, we consider 12 distinct cases for latency and device constraints. Upon each training, our method generates 200 architectures which are then queried on the benchmark. We adopt two conditions simultaneously, namely the accuracy should be in $> f_{th}(= 95)$ percentile class and the latency should satisfy the given constraint. We repeat the generation process for 10 runs and report the mean of the validation accuracy, along with the feasibility and number of queries.

**ImageNet** We start by generating 100 architectures through our approach trained on NAS-Bench-301. Then, we select the best architecture in terms of validation accuracy. Next, we train the network using the same training pipeline and code as TENAS (Chen et al., 2021a). Finally, we save the weights from the epoch where the top-1 and top-5 validation errors are minimum and report the top-1 and top-5 errors in Table 5.

## A.6 Benchmark Descriptions

### A.6.1 NAS-Bench-101

NAS-Bench-101 (Ying et al., 2019) is a cell-based tabular benchmark, comprising a large collection of 423,624 distinct architectures represented as cells. These architectures are also mapped to their respective validation and test accuracy metrics, evaluated on CIFAR-10 image classification task. In this benchmark, the cells

are constrained to have a maximum of 7 nodes and 9 edges. Specifically, the first and last nodes within these cells serve as input and output nodes. Intermediate nodes within the cells can take on one of three possible operations: 1x1 convolution, 3x3 convolution, or 3x3 max-pooling. Furthermore, it is important to note that each convolutional operation is preceded by batch normalisation, followed by a Rectified Linear Unit (ReLU) activation function.

### A.6.2 NAS-Bench-201

Another cell-based tabular benchmark is NAS-Bench-201 (Dong & Yang, 2020), which contains data for 15,625 architectures (cells) trained on 3 datasets- CIFAR-10, CIFAR-100 (Krizhevsky et al., 2009) and ImageNet16-120 (Deng et al., 2009). In contrast to NAS-Bench-101, each edge of a cell in NAS-Bench-201 is associated to an operation drawn from a predefined operation set $\mathcal{O} = \{$1x1 convolution, 3x3 convolution, 3x3 avg pooling, skip, zero$\}$. In our training and experiments on NAS-Bench-201, we convert the edge based representation to node-based representation, where each node is associated with an operation, similar to NAS-Bench-101. This conversion is in line with the conversion in Arch2Vec (Yan et al., 2020). Each cell comprises 4 nodes and 6 edges and the adjacency matrices are identical to one another. The existence of operations like zero and skip enforces the structural diversity in different architectures.

### A.6.3 NAS-Bench-301

NAS-Bench-301 (Siems et al., 2021) is a surrogate benchmark that trains and evaluates several performance predictors on 60,000 sampled architectures from DARTS search space (Liu et al., 2019). These learned performance (surrogate) predictors are then able to predict the accuracy of architectures in DARTS search space (comprising $10^{18}$ architectures). The architectures in DARTS comprise of a normal cell and a reduction cell. Each cell has a maximum of 7 nodes and 12 edges with each edge associated with an operation drawn from the set $\mathcal{O} = \{$ 3x3 sep. conv., 3x3 dil. conv., 5x5 sep. conv, 3x3 average pooling, identity, zero $\}$. We utilise a pretrained XGBoost (Chen & Guestrin, 2016) provided by Siems et al. (2021) as the surrogate predictor for our experiments.

### A.6.4 NAS-Bench-NLP

NAS-Bench-NLP is the first NAS benchmark designed for Natural Language Processing tasks (Klyuchnikov et al., 2020). While its search space is extremely large with the total of $10^{53}$ architectures, NAS-Bench-NLP provides 14,322 architectures trained on Penn TreeBank dataset (Marcus et al., 1993). Each cell in the search space has a maximum of 24 nodes, 3 hidden states and 3 linear input vectors. The nodes in each cell depict the operations drawn from the set $\mathcal{O} = \{$Linear, blending, product, sum, tanh, sigmoid, LeakyRELU $\}$.We utilise the surrogate predictor provided by NAS-Bench-X11 (Yan et al., 2021a) for this benchmark.

### A.6.5 HW-NAS-Bench

HW-NAS-Bench is a unique benchmark that provides hardware-specific details, including latency and energy cost, across various devices along with their respective accuracy. These devices encompass a diverse set of hardware platforms, including EdgeGPU, Raspi4, Pixel3, EdgeTPU, Eyeriss, Pixel3, and FPGA. Crucially, HW-NAS-Bench operates within two distinct search spaces: NAS-Bench-201 (Dong & Yang, 2020) and FB-Net (Wu et al., 2019). In our experiments, we utilise latency information as hardware constraint, within the context of the NAS-Bench-201 search space.

## A.7 Examples of generated cells

Here, we demonstrate some examples of the generated normal cells on DARTS search space using our proposed method trained on NAS-Bench-301 (Siems et al., 2021) (demonstrated in figure 2) and indicate structural differences compared to cells generated by DARTS (Liu et al., 2019) and TENAS (Chen et al., 2021a) (demonstrated in figure 3). It can be observed that DiNAS generations have a marginally higher 5x5 convolutional connections compared to the other approaches.

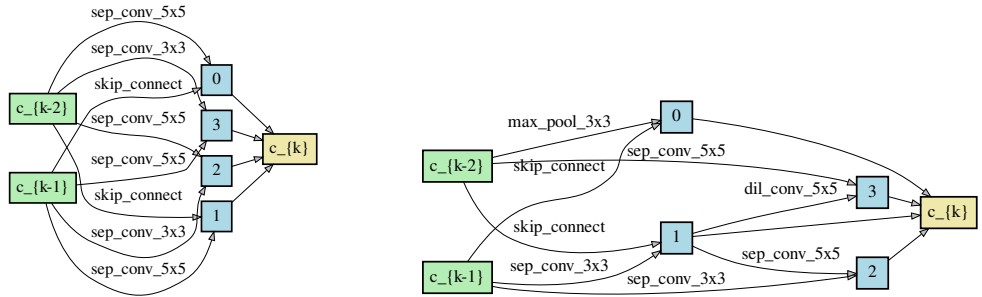

Figure 2: Examples of high performing generated cells by our method DiNAS on DARTS search space using NAS-Bench-301.

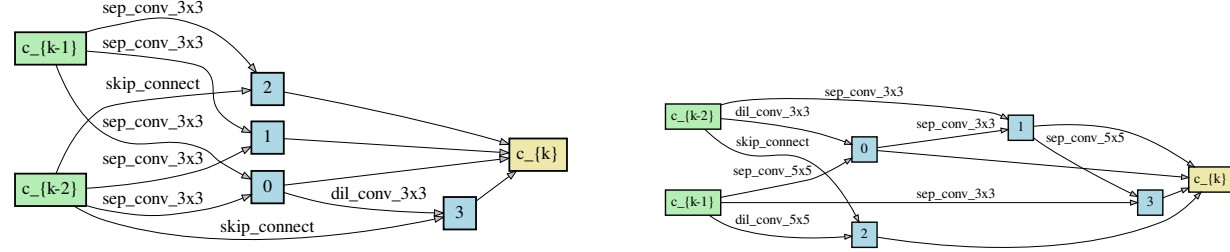

Figure 3: Examples of high performing generated cells by DARTS (Liu et al., 2019) (left) and TENAS (Chen et al., 2021a) (right)

