# OpenReview forum: "Multi-conditioned Graph Diffusion for Neural Architecture Search"
_TMLR — Accepted by TMLR_

### Review · Reviewer_uVxU · 2023-12-21

**Summary Of Contributions:**

The paper proposes a neural architecture search (NAS) approach called Multi-conditioned Graph Diffusion for NAS (DiNAS). It uses discrete conditional graph diffusion processes to generate high-performing neural network architectures. It introduces a multi-conditioned classifier-free guidance technique to impose multiple constraints (e.g. accuracy, latency) jointly when generating architectures. This allows searching for architectures that meet multiple objectives. The method is completely differentiable, requiring only a single model training. It achieves promising results on six NAS benchmarks, yielding novel architectures very quickly (less than 0.2 seconds per architecture). Experiments demonstrate the approach generates novel, unique architectures. The method also generalizes well, as shown by experiments on ImageNet.

**Audience:**

Yes

**Claims And Evidence:**

Yes

**Requested Changes:**

See weaknesses above.

**Strengths And Weaknesses:**

### Strengths

* Single model training is more efficient than methods requiring separate predictor networks. Most generative NAS methods need an additional predictor model to estimate performance of generated architectures. Training two models is less efficient. By using classifier-free guidance, this method trains only the generator model itself, reducing total training costs.

* Multi-conditioned guidance technique is flexible for imposing multiple constraints, an important capability. Guiding the search with multiple objectives like accuracy and efficiency is crucial for real applications but not well supported in previous generative NAS techniques. The proposed multi-conditioned guidance provides an elegant way to flexibly impose multiple conditions, enabling robust multi-objective search.

* Fast architecture generation rate enables rapid exploration of broad search spaces. Generating just under 0.2 seconds per architecture is orders of magnitude faster than previous approaches that require full architecture evaluation. This rapid rate allows searching much larger spaces, finding high-performing architectures faster than other methods.

* Achieves state-of-the-art or comparable results across several NAS benchmark tasks. Solid performance matching or exceeding prior state-of-the-art on major NAS benchmarks demonstrates this method's effectiveness. Competitive accuracy to more complex techniques shows the power of the proposed graph diffusion approach.

### Weaknesses

* The method does not always achieve absolute state-of-the-art accuracy, slightly underperforms some other methods. While overall results are highly competitive, top accuracies fall just short of 1-2 other approaches in some benchmarks. There may still be room to improve architecture quality further through guidance tuning or alternate diffusion formulations.

* Choice of guidance scale parameter affects diversity/quality tradeoff and the optimal values are likely dataset-specific. The hyperparameter $\gamma$ balances diversity and quality during sampling. Best setting probably depends greatly on factors like search space size and conditioning constraints. More analysis could better characterize how to set $\gamma$ robustly.

* The top-1/5 errors on ImageNet are slightly higher than some strong baselines. Specifically, ImageNet results were not superior to all baselines as with CIFAR experiments. Factors such as architecture conditioning strengths may differ when generalizing to very large datasets.

* Does not provide intuitive case studies to fully demonstrate advantages. While numerical results show accuracy and efficiency gains, the paper lacks detailed case studies to more intuitively showcase the benefits versus other methods. Providing examples that compare architectures found across different search techniques could give better insights into real-world improvements. More qualitative analysis would strengthen overall motivation for the proposed approach.

---

> ### Author Response · Authors · 2024-01-22
> **Rebuttal by Authors for Reviewer uVxU (1/1)**
>
> We are sincerely thankful to the reviewer for the insightful comments and valuable suggestions to our paper. Below, we address the reviewer's concerns:
>
> > 1. The method does not always achieve absolute state-of-the-art accuracy, slightly underperforms some other methods ... There may still be room to improve architecture quality further through guidance tuning or alternate diffusion formulations.
> > 2. Choice of guidance scale parameter affects diversity/quality tradeoff and the optimal values are likely dataset-specific. The hyperparameter balances diversity and quality during sampling. Best setting probably depends greatly on factors like search space size and conditioning constraints. More analysis could better characterize how to set  robustly.
>
> - Upon the reviewer's recommendation, we have performed an ablation study comparing the performance of the model on two differently sized search spaces NAS-Bench-101 (size of 423k samples) and NAS-Bench-NLP (size of $10^{53}$ samples) search space when using different guidance scales. The results are as follows:
>
> **Table: Comparison of performance of DiNAS using different guidance scales on NAS-Bench-101 and NAS-Bench-NLP search spaces.**
> | $\gamma$ | NAS-Bench-101 |NAS-Bench-101| NAS-bench-NLP
> | ---|--|--|--|
> | | **Val acc.** | **Test acc.** |**Val acc.**
> |-4| 94.98+-0.17|**94.27+-0.20**|**96.06+-0.17**|
> |-2|**95.06+-0.14**|94.16+-0.20|95.97+-0.12|
> |2|92.60+-0.43|92.05+-0.58|95.42+-0.08|
> |4|91.44+-0.47|90.86+-0.59|95.36+-0.29|
>
> - We observe from the table that negative values of guidance scale perform better than positive values, unlike [1]. This can be attributed to the difference in formulation of the guidance parameter, which is $w$ in [1], instead of $\gamma$ ($w \sim -\gamma$).
> - We found that $\gamma=-4$ setting produces best results for both, small and large search spaces, and is thus used in our experiments.
> - We have updated the paper with this ablation study in Appendix Section A.4.1
>
> > 3. The top-1/5 errors on ImageNet are slightly higher than some strong baselines. Specifically, ImageNet results were not superior to all baselines as with CIFAR experiments. Factors such as architecture conditioning strengths may differ when generalizing to very large datasets.
>
>
> - Thank you for raising this point. The comparison of our approach to non-generative methods gives our approach a disadvantage due to the unavailability of a dataset with DARTS-style normal-reduced cell architectures and ImageNet accuracies. Although the transferability works well in this case, it does not beat the state-of-the-art (by non-generative method DrNAS) unfortunately. Moreover, unlike some other generative NAS methods, e.g. [2], we conduct ImageNet experiments along with other benchmarks. Moreover,
> - Comparing to AG-Net [3], the reason for their better performance can be attributed to the extra Latent Space Optimization (LSO) technique, which our approach omits to improve the search cost (see Table 6).
> - We have included this explanation in our updated manuscript as well (Section 5.2).
>
> >4. Does not provide intuitive case studies to fully demonstrate advantages. ... Providing examples that compare architectures found across different search techniques could give better insights into real-world improvements. More qualitative analysis would strengthen overall motivation for the proposed approach.
>
> -  After a visual analysis, we have observed that DiNAS generations have a marginally higher 5x5 convolutional connections compared to the other approaches.
> -  We have updated our transcript with the examples of our found architectures and indicated the important structural differences when compared to other recent approaches in the Appendix section A.7.
>
> References
>
> [1] Ho, J., & Salimans, T. (2022). Classifier-free diffusion guidance. arXiv preprint arXiv:2207.12598.
>
> [2] An, S., Lee, H., Jo, J., Lee, S., & Hwang, S. J. (2023). DiffusionNAG: Task-guided Neural Architecture Generation with Diffusion Models. arXiv preprint arXiv:2305.16943.
>
>  [3] Lukasik, J., Jung, S., & Keuper, M. (2022, October). Learning where to look–generative nas is surprisingly efficient. In European Conference on Computer Vision (pp. 257-273). Cham: Springer Nature Switzerland.

---

### Review · Reviewer_kwHT · 2023-12-26

**Summary Of Contributions:**

The paper addresses the task of neural architecture search and proposed a graph-diffusion approach. Existing techniques from the literature are employed for performing discrete conditional graph diffusion. The paper introduces an extension involving multiple conditions to encourage satisfaction of multiple constraints such as accuracy and latency. The proposed approach is differentiable and thus permits end-to-end training and only requires the training of a single model. The paper reports the results of experiments on commonly studied benchmarks and shows that the proposed approach offers an attractive trade-off between speed and accuracy.

**Audience:**

Yes

**Claims And Evidence:**

No

**Requested Changes:**

1.	The authors mention the papers by An et al. (2023) and Lukasik et al. (2022) in the introduction but An et al. (2023) is not discussed in the Related Work section and Lukasik et al. (2022) is only mentioned in passing. These seem to be the most closely related works, so the relationship with these works should be discussed in detail. In particular, the paper claims in the introduction that because the approach “is completely differentiable and thus training involves only a single model … we reach promising performance with much smaller search time.” The paper demonstrates this to some degree for AG-Net (although I think the reporting of timing results needs to be much clearer, as requested below). However, I do not believe there is any experimental comparison to the method proposed by An et al. (2023). This is a recent work, so there could be an argument that it is concurrent research. On the other hand, the paper by An et al. (2023) was posted on arXiv in May, and that is the version that the authors are citing, so presumably they were aware of the work. There are comparable experimental results that could be taken from An et al. (2023), even if the experiments were not reproduced. Moreover, if there is no experimental comparison, the paper should not claim that there is a “much smaller search time”. It should be made clear that this claim refers only to Lukasik et al. (2022). In general, I would like to see a considerably more detailed discussion of why the authors believe that an approach involving a single model is so important.

2.	Tables 1 and 2 report only the number of queries. Are the search times directly proportional to the number of queries? Or does it vary substantially for different methods? In particular, does the proposed diffusion approach require significantly more time per query?

3.	The paper does not provide any intervals or deviations for the averaged results. The reader would have better insight into the performance of the method if such information were included. For example, in Table 2, what is the variability over different runs for ImageNet for both the validation and test sets? (Perhaps this would explain

4.	With regard to the results in Table 2, can the authors provide a more detailed explanation of
why there is a significant discrepancy between the validation and test results for ImageNet? The current explanation is that the proposed approach optimizes over validation accuracy and this does not necessarily correlate with test performance. While this is true, it is odd that the only technique that is affected in a significant way is the proposed method. The variability over different runs would be helpful here – is the proposed method identifying exactly the same architecture, irrespective of instantiation? Or is one of the high-scoring validation architectures particularly poor in the test set?

5.	Table 5 presents important ImageNet experimental results. However, the selection of the baselines is surprising, with only AG-Net being a recent comparable baseline. There are multiple other recent baselines that could be included (if only to provide the reader with a better understanding of the competitive landscape). Some suggestions might be [R1] (although this is a very recent publication, it was posted on arXiv in 2022), [R2] and [R3]. There is also [R4], but this is a very recent publication, so it is perhaps unfair to ask for inclusion/comparison – perhaps the authors could at least cite and discuss. Arch2VEC (Yan et al., 2020) [R5] and DrNet (Chen et al., 2021) [R6] are also relevant baselines and should be cited and discussed (possibly with some experimental comparison).

6.	In Table 5, why is the training time not included in the execution time? The paper refers to this as a “one-time cost”. For a fair comparison, it would seem more reasonable just to have a column that reports the total execution time. (This can be broken down in other columns to provide more information about how the time is being used). Is there a demonstration of transferability to multiple searchers if this the training time is to be considered a “one-time cost”. (I struggled to understand this from the paper – perhaps the authors could add a sentence or two to highlight how the training applies to multiple different searches). At the moment, the value for T1 isn’t even reported, making it very hard to interpret this table and compare the times for different architectures. This is important, because a key claim of the paper is that the method is faster than AG-Net (the accuracy performance in Table 5 is inferior).

7.	There is a concern that the metrics for DARTS, TE-NAS, NASNET-A were all taken from respective papers. This means that only AG-Net follows the protocol specified in the paper. Since the gpus are different and the experimental procedure has changed, it seems unreasonable to have a table that reports numbers as though they can be directly compared.

[R1] Yang, Longxing, et al. "Sweet Gradient matters: Designing consistent and efficient estimator for Zero-shot Architecture Search." Neural Networks 168 (2023): 237-255.

[R2] Ye, Peng, et al. "b-darts: Beta-decay regularization for differentiable architecture search." Proceedings of the IEEE/CVF Conference on Computer Vision and Pattern Recognition. 2022.

[R3] Soro, Bedionita, and Chong Song. "Enhancing Differentiable Architecture Search: A Study on Small Number of Cell Blocks in the Search Stage, and Important Branches-Based Cells Selection." Proceedings of the IEEE/CVF International Conference on Computer Vision. 2023.

[R4] Li, Muchen, et al. "GraphPNAS: Learning Probabilistic Graph Generators for Neural Architecture Search." Transactions on Machine Learning Research (2023).

[R5] Shen Yan, Yu Zheng, Wei Ao, Xiao Zeng, and Mi Zhang. Does unsupervised architecture representation learning help neural architecture search? In Advances in Neural Information Processing Systems (NeurIPS), 33:12486– 12498, 2020.

[R6] Xiangning Chen, Ruochen Wang, Minhao Cheng, Xiaocheng Tang, and Cho-Jui Hsieh. Dr{nas}: Dirichlet neural architecture search. In Proc. International Conference on Learning Representations, 2021.

**Strengths And Weaknesses:**

Strengths
1.	Although there have been some very recent methods proposing diffusion approaches for network architecture search, these are contemporaneous with the submitted paper. The novel approach will be of significant interest to multiple researchers who are exploring the topic of network architecture search.
2.	The proposed method appears to offer advantageous performance (although there is a need for clearer reporting of execution time to better support some of the main claims of the paper).
3.	The methodology is clearly described and the experiments are thorough. The reporting of novelty/uniqueness is a particularly welcome and interesting inclusion.

Weaknesses
1.	Some closely related work is not discussed in detail. Although some of this related work is very recent, it is cited in the paper, so the authors were clearly aware of it. While it may be unreasonable to expect a detailed experimental comparison, a more thorough qualitative discussion is important, and if experimental results are available, then it seems reasonable to include them in the tables.
2.	The results in the paper do not include any indication of variability over different runs.
3.	In some cases, the choice of baselines could be improved. There is a multitude of recent techniques – some of the studies compare to only one recent method.
4.	The reported execution times are difficult to understand, making it challenging to compare the proposed method with the baselines. This is important, considering that a major claim of the paper is that the required time is dramtically reduced.

---

> ### Author Response · Authors · 2024-01-22
> **Rebuttal by Authors for Reviewer kwHT (1/3)**
>
> We would like to express our sincere gratitude to the reviewer for the comprehensive and detailed review, aimed at increasing the quality and robustness of our paper. To address the reviewer's concerns, we aim to provide clarification on each point raised in a sequential manner.
>
> > 1. The authors mention the papers by An et al. (2023) and Lukasik et al. (2022) in the introduction but An et al. (2023) is not discussed in the Related Work section and Lukasik et al. (2022) is only mentioned in passing ... detailed discussion of why the authors believe that an approach involving a single model is so important.
>
> - We have performed qualitative and quantitative comparisons with the work of [1] and [2], including experimental performance and search time comparisons with [1] on CIFAR-100 (see table below).
>
> **Table: Search and training time comparisons with DiffusionNAG [1]**
> | Method | Search/Generation Time (sec)| Training time (hrs)
> | -------- | ----| ---- |
> |DiffusionNAG [1]| 261| 3.43
>  **DiNAS (ours)**  |  **15.36** | **0.25**  |
>
> - We observe that our approach generates architectures faster than DiffusionNAG [1] by a significant margin. It can thus be proved that our approach, using diffusion without an external classifier, is more efficient than the DiffusionNAG, due to its reduced computational requirement for each generation.
> - We have modified our manuscript with the updated results, observed in Tables 2 and 6 and the qualitative comparison, observed in Section 2.
>
> > 2. Tables 1 and 2 report only the number of queries. Are the search times directly proportional to the number of queries? ...
>
> - We would like to point out that we follow the same evaluation protocol and compare with the same evaluation metrics used in previous work  [2] [3] [4]. Although there are different other factors (like the nature of the algorithm) affecting the search time for different approaches, generally search times are directly proportional to the number of queries, and is thus used as an efficiency metric by previous approaches [2][3][4].
> - We have updated our manuscript to make this point clearer (Section 5.1).
>
> >  In particular, does the proposed diffusion approach require significantly more time per query?
> - Our diffusion approach requires insignificant time for querying (less than 0.2 seconds per architecture) because we use the learned latent space to generate architectures.
>
> >3. The paper does not provide any intervals or deviations for the averaged results ...
>
> -  Considering the reviewers valuable recommendation, we report the standard deviation (StD) along with the mean of our results on various benchmarks.
> -  We observe, from the table below, that standard deviation (StD.) for test accuracy is generally higher than the StD. for validation accuracy, particularly for NAS-Bench-201 on ImageNet, because the model is trained to generate architectures with high validation accuracy and has no test accuracy information.
>
> **Table: Results including Standard Deviation (StD)**
> | Experiment | Dataset| Val. acc. (Mean +- StD.) | Test Acc. (Mean+- StD.)
> | -------- | ----| ---- | ---- |
> |NAS-Bench-101 | CIFAR-10 | 94.98 +-0.169 | 94.27+-0.197
> |NAS-Bench-201 | CIFAR-10 | 91.61 +- 0.000 | 94.37 +- 0.184
> |NAS-Bench-201 | CIFAR-100 | 73.49 +- 0.000 | 73.51 +- 0.000
> |NAS-Bench-201 | ImageNet | 46.66 +- 0.092 | 45.41 +- 0.589
> |NAS-Bench-301 | CIFAR-10 | 94.92 +- 0.072 | - |
> |NAS-Bench-NLP | Penn Tree Bank | 96.06 +- 0.173 | -
>
> - We have now included the standard deviation in all the experiments performed in the appendix (Appendix Section A.3).
>
> > 4. With regard to the results in Table 2, can the authors provide a more detailed explanation of why there is a significant discrepancy between the validation and test results for ImageNet? ... is the proposed method identifying exactly the same architecture, irrespective of instantiation? Or is one of the high-scoring validation architectures particularly poor in the test set?
>
> -  The reason for this discrepancy is that through experimentation, we found that some of the generated high validation accuracy architectures for the case of Nas-Bench-201-ImageNet do not have a high test accuracy. (one of the architectures has a val. acc. of 46.73% while a test acc. of 44.63% while another has a val. acc. of 46.73% and a test acc of 44.81%). However, because there are multiple high performing architectures based on the validation accuracy, some of them (at least one) do have the highest test accuracy. As the test accuracy is completely unknown to the model, its just a matter of luck that the generation will have the highest test accuracy as well.
> -  This explanation is provided in the updated manuscript (Section 5.2). We can also observe this discrepancy in the standard deviation (reported in Table 10) corresponding to the test accuracy for ImageNet.

---

> ### Author Response · Authors · 2024-01-22
> **Rebuttal by Authors for Reviewer kwHT (2/3)**
>
> >5. Table 5 presents important ImageNet experimental results. However, the selection of the baselines is surprising, with only AG-Net being a recent comparable baseline ...
>
> - We thank the reviewer for the suggestions for the baselines. We have added the suggested baselines, e.g. DrNAS [5] (2020), Sweetimator [6] (2023), DARTS++ [7] (2023) and $\beta$-DARTS [8] (2022), in our ImageNet experiments.
>
> >6. In Table 5, why is the training time not included in the execution time? ... This is important, because a key claim of the paper is that the method is faster than AG-Net.
>
> -  Our search times reporting is concurrent with the protocols in [2], [3], [4]. The training time is a one-time cost as the trained model can be leveraged to generate infinite architectures. Since the search time refers to the duration needed to search for a single architecture, following the model's pre-training, our search cost is solely the inference cost. Thus, our table states the search time and one-time training costs seperately. The search process can be iterated within the same search space, and our approach will consistently incur just the inference cost.
> -  We have restructured the Table 5, splitting the original table into two seperate tables (Tables 5 and 6) (also provided below) in the updated manuscript to enhance clarity. We have also provided further explanation of the numbers in table 6, mainly explaining why the training time is considered as a one-time cost (Section 5.3.1).
>
> **Table: Comparison of results for top-1, top-5 errors on ImageNet.**
> | Methods| Top-1 $\downarrow$ | Top-5 $\downarrow$ | Queries$\downarrow$
> |---|----|----|------|
> | NASNET-A  | 26.0  | 8.4 | 20000  |
> | DARTS  | 26.7  | 8.7| -  |
> | DrNAS | 23.7   | 7.1 | -   | -   |
> | TENAS  | 26.2               | 8.3  | -                   | 1.2  |
> | AG-Net  | 24.1               | 7.2  | 304 |
> | Sweetimator        | 24.1  | -  | -  | 0.192  |
> | DARTS$^{++}_{aug}$ | 24.8| 7.8| - | -  |
> | $\beta$-DARTS      | 23.9  | 7.0       | -     | -    |
> | DiNAS (ours) | 24.8  | 7.4| 100
>
>
> **Table: Comparison of search times in GPU seconds and training times of the pre-trained generator in GPU hours on ImageNet and CIFAR 100 (using NAS-Bench-201) for different approaches.**
>
> | Dataset |Methods |Search/Gen. Time (GPU sec.)$\downarrow$ | Pre-training cost (GPU hrs) $\downarrow$ |
> |---------------------------|--------------------------------------|-----------------------------------------------------------|------------------------------------------------------------|
> | ImageNet| NASNET-A  | 1.7x $10^8$  | -      |
> | ImageNet | DARTS  | 345600   | -  |
> | ImageNet| DrNAS | 397440  | -   |
> | ImageNet | TENAS  | 4320  | -   |
> | ImageNet | AG-Net  | 1728  | 21.6 |
> | ImageNet  | $\beta$-DARTS      | 34560  | -  |
> | ImageNet| DARTS$^{++}_{aug}$ | 25920 | - |
> | CIFAR100  | DiffusionNAG| 261 | 3.43  |
> | ImageNet | **DiNAS (ours)** | 53.76   | 16.6
> | CIFAR100 | **DiNAS (ours)** | **15.36**   | **0.25**  |
>
> -   Moreover, we show using experiments on ImageNet (Table 5) that our model trained on one dataset (e.g. CIFAR-10) can generate architectures that can be generalized to different datasets (e.g. ImageNet).

---

> ### Author Response · Authors · 2024-01-22
> **Rebuttal by Authors for Reviewer kwHT (3/3)**
>
> >7. There is a concern that the metrics for DARTS, TE-NAS, NASNET-A were all taken from respective papers. This means that only AG-Net follows the protocol specified in the paper...
>
> - We would like to point out that we use the same ImageNet training protocol as [9] and [2] which is available on github [10].
> >8. Since the gpus are different and the experimental procedure has changed, it seems unreasonable to have a table that reports numbers as though they can be directly compared.
>
> - The reviewer's observation regarding the impracticality of comparing search times across different GPUs is duly acknowledged. However, the closest comparison that we follow, which is also followed by most of the relevant literature [5], [6], [11] , is to use GPU hours/days as a metric of comparison. This eliminates the variability in using multiple GPUs and GPU clusters for our task.
>
> References
>
> [1] An, S., Lee, H., Jo, J., Lee, S., & Hwang, S. J. (2023). DiffusionNAG: Task-guided Neural Architecture Generation with Diffusion Models. arXiv preprint arXiv:2305.16943.
>
> [2] Lukasik, J., Jung, S., & Keuper, M. (2022, October). Learning where to look–generative nas is surprisingly efficient. In European Conference on Computer Vision (pp. 257-273). Cham: Springer Nature Switzerland.
>
> [3] Yan, S., Zheng, Y., Ao, W., Zeng, X., & Zhang, M. (2020). Does unsupervised architecture representation learning help neural architecture search?. Advances in neural information processing systems, 33, 12486-12498.
>
> [4] Wu, J., Dai, X., Chen, D., Chen, Y., Liu, M., Yu, Y., ... & Yuan, L. (2021). Stronger nas with weaker predictors. Advances in Neural Information Processing Systems, 34, 28904-28918.
>
> [5] Chen, X., Wang, R., Cheng, M., Tang, X., & Hsieh, C. J. (2020). Drnas: Dirichlet neural architecture search. arXiv preprint arXiv:2006.10355.
>
> [6] Longxing Yang, Yanxin Fu, Shun Lu, Zihao Sun, Jilin Mei, Wenxiao Zhao, Yu Hu,Sweet Gradient matters: Designing consistent and efficient estimator for Zero-shot Architecture Search, Neural Networks,Volume 168, 2023,Pages 237-255, ISSN 0893-6080
>
> [7] Soro, Bedionita, and Chong Song. "Enhancing Differentiable Architecture Search: A Study on Small Number of Cell Blocks in the Search Stage, and Important Branches-Based Cells Selection." Proceedings of the IEEE/CVF International Conference on Computer Vision. 2023.
>
> [8]  Ye, Peng, et al. "b-darts: Beta-decay regularization for differentiable architecture search." Proceedings of the IEEE/CVF Conference on Computer Vision and Pattern Recognition. 2022.
>
> [9] Chen, W., Gong, X., & Wang, Z. (2021). Neural architecture search on imagenet in four gpu hours: A theoretically inspired perspective. arXiv preprint arXiv:2102.11535.
>
> [10] DARTS evaluation (https://github.com/chenwydj/DARTS_evaluation)
>
> [11] Liu, H., Simonyan, K., & Yang, Y. (2018). Darts: Differentiable architecture search. arXiv preprint arXiv:1806.09055.

---

### Review · Reviewer_3zRw · 2024-01-11

**Summary Of Contributions:**

The paper presents a diffusion based generative model for neural architecture search. The architectures are represented as a graph, a diffusion model is trained over the distribution of nodes and edges of graphs corresponding to architectures. The proposed diffusion models is a small variation of an existing diffusion model over graphs. The trained diffusion model can be used in a search process to find models with high accuracy.

Extensive experiments are conducted on multiple tabular NASBench datasets, and ImageNet. The proposed approach offers marginal benefits over prior work.

**Audience:**

Yes

**Broader Impact Concerns:**

There are no perceived ethical concerns for this work.

**Claims And Evidence:**

No

**Requested Changes:**

- Please state the concrete claims being made by the paper and what evidence (if any) has been provided to validate the claims. The claims are not apparent from the paper.
- For the novelty and uniqueness of architectures, how was the similarity of a pair of generated graphs measured? Was this through a WL-test? This information is missing.
- How would the novelty and uniqueness vary as you sample lot more architectures? I would presume, there are only a small set of “good” architectures, and once those are sampled, you will start seeing lots of repetitions.
- For the results, can you also provide standard deviations of accuracy? Only mean values have been provided. Same for the number of queries (if it changes from one run to another).
- In section 4.1.2, what is $a$? It has not been defined as far as I can tell.
- The purpose of discretization of the target variable in section 4.1.3 is unclear. As in, why do we need this discretization? Through discretization the diffusion model will have to learn the density over regions with low accuracy networks. Almost always, our goal is to maximize accuracy. So in this case, isn’t it sufficient to learn the density over networks above a certain accuracy threshold and ignore the rest? This goes back to the weakness mentioned above, in terms of, lack of clarity on what limitations of existing density models is the proposed approach designed to overcome?

**Strengths And Weaknesses:**

Strengths:
- The proposed approach shows marginal improvements over prior work on multiple NAS benchmarks.
- The novelty and uniqueness analysis is interesting.

Weaknesses:
- The primary weakness is the lack of a concrete claim in this paper. It is unclear to this reviewer why we need to learn the density of architectures using a diffusion model. What conceptual or practical limitation of existing density models for the same (e.g., VAE or its variants) is the proposed approach addressing.
- The multi-condition idea is interesting in principle. However, the particular multi-condition used here in terms of different accuracy quantiles is strange. Under what circumstances should quantiles corresponding to low accuracy of interest? Wouldn’t we always want models in the top say 95th percentile?

---

> ### Author Response · Authors · 2024-01-22
> **Rebuttal by Authors for Reviewer 3zRw (1/2)**
>
> We extend our sincere appreciation to the reviewer for their valuable suggestions and insightful comments. Below, we address the concerns of the review.
>
> > 1. It is unclear to this reviewer why we need to learn the density of architectures using a diffusion model. What conceptual or practical limitation of existing density models for the same (e.g., VAE or its variants) is the proposed approach addressing?
>
> - To the best of our knowledge, the previous generative NAS methods either use a supernet [1] , a simple GAN style generator [2] or use a classifier-based diffusion model [3] for NAS, which have their own limitations. In particular, [1] requires training an expensive supernet, [2] requires an external predictor for a slower node-by-node generation, and [3] requires an additional classifier that incorporates noisy data for classification, which theoretically hurts the performance of the diffusion model and is slower [4].
> - We have made this point clear in our updated manuscript in Related work section as well.
> - The choice of diffusion models comes from the fact that diffusion models offers precise control in generation compared to previous methods like GANs or VAEs. This helps us guide the model step by step towards specific goals, like  high accuracy and low latency. Past studies, like [5], have proven that having this kind of control across different types of data helps improve the generation quality.
> > 2. Under what circumstances should quantiles corresponding to low accuracy of interest? Wouldn’t we always want models in the top say 95th percentile?
>
> - We would like to clarify that given the top 5% of architectures exhibit a notably low sample count, the diffusion model would not be able to learn the structure of the valid architectures if it is solely trained on the top 5% architectures. Since it is required to label all the architectures in the training set, we have to assign a class to low accuracy architectures as well.
> - We have made this point clear in our updated manuscript (Section 5.1-Implementation)
>
> >3. Please state the concrete claims being made by the paper and what evidence (if any) has been provided to validate the claims.
>
> - We would like to make clear that the concrete claim of this paper, which we emperically prove, is that guided graph diffusion, specifically discrete graph diffusion with multi-conditioned classifier free guidance based NAS approach, should work better than previous generative and traditional NAS methods due to the model's ability to perform architecture generation in a controlled guided manner without the need of an external predictor.
> - We have improved the clarity of the claims made in our updated manuscript (Section 1).
>
>
> > 2. For the novelty and uniqueness of architectures, how was the similarity of a pair of generated graphs measured? Was this through a WL-test?
>
> -  Our protocol for evaluating novelty and uniqueness is concurrent with [2], [3], and [6].
> - We do not use a WL-Test- The assessment is rather simpler. To assess novelty, we calculate the percentage of generated samples absent in the training data whereas to assess uniqueness, we calculate the ratio of architectures present just once in the generations to the total number of generations.
>
> > 3. How would the novelty and uniqueness vary as you sample lot more architectures? I would presume, there are only a small set of “good” architectures, and once those are sampled, you will start seeing lots of repetitions.
>
>
> - Upon the reviewer's concern, we have added some additional results to our ablation study, mainly concerning when a large number of architectures (100,000) is sampled.
>
> **Table: Novelty and Uniqueness Analysis**
> | Benchmark | Generations| Novelty (%) | Uniqueness(%)
> | -------- | ----| ---- | ---- |
> |NAS-Bench-101 | 100,000 | 100| 91.10
> |NAS-Bench-NLP | 100,000 | 100 | 96.78
> |NAS-Bench-101 | 2,000 | 100| 97.37
> |NAS-Bench-NLP | 2,000 | 100 | 97.57
>
> - We observe that while the novelty for both the cases (2000 and 100,000) remain at the top, the uniqueness suffers a bit when sampling high number of architectures, due to a limited number of possible high-performing architectures.
> - We have integrated the study in the Section 5.3.2 of the updated manuscript.

---

> ### Author Response · Authors · 2024-01-22
> **Rebuttal by Authors for Reviewer 3zRw (2/2)**
>
> > 4. For the results, can you also provide standard deviations of accuracy?
>
> -  In response to the insightful recommendations from the reviewers, we have incorporated the standard deviation (StD) along with the mean for our results across diverse benchmarks.
> -  We observe, from the table below, that standard deviation (StD.) for test accuracy is generally higher than the StD. for validation accuracy, particularly for NAS-Bench-201 on ImageNet, because the model is trained to generate architectures with high validation accuracy and has no test accuracy information.
>
> **Table: Results including the standard deviation (StD)**
> | Experiment | Dataset| Val. acc. (Mean +- StD.) | Test Acc. (Mean+- StD.)
> | -------- | ----| ---- | ---- |
> |NAS-Bench-101 | CIFAR-10 | 94.98 +-0.169 | 94.27+-0.197
> |NAS-Bench-201 | CIFAR-10 | 91.61 +- 0.000 | 94.37 +- 0.184
> |NAS-Bench-201 | CIFAR-100 | 73.49 +- 0.000 | 73.51 +- 0.000
> |NAS-Bench-201 | ImageNet | 46.66 +- 0.092 | 45.41 +- 0.589
> |NAS-Bench-301 | CIFAR-10 | 94.92 +- 0.072 | - |
> |NAS-Bench-NLP | Penn Tree Bank | 96.06 +- 0.173 | -
>
> - We have now included the standard deviation in all the experiments performed in the appendix (Appendix Section A.3).
>
> - We have now included the standard deviation of all results involving multiple runs in the appendix section A.3.
>
>
> - The number of queries stay the same for all the runs of an experiment.
>
>  > 5. In section 4.1.2, what is ***a***. It has not been defined as far as I can tell.
>
> - We apologize for the incorrect notation. *a* here refers to the performance metric or the target variable (e.g. accuracy or latency). For simplicity, we have replaced *a* with *y* in our updated manuscript.
>
> >6. The purpose of discretization of the target variable in section 4.1.3 is unclear. As in, why do we need this discretization? ... isn’t it sufficient to learn the density over networks above a certain accuracy threshold and ignore the rest?
>
> - Since, the top 5% of architectures or so comprise of very low number of samples, the diffusion model would not be able to learn the structure of the valid architectures if it is just trained on top 5% architectures.
>
> - We have made the necessity of discretization clear in the manuscript as well.
>
> References
>
> [1] Huang, S. Y., & Chu, W. T. (2021). Searching by generating: Flexible and efficient one-shot NAS with architecture generator. In Proceedings of the IEEE/CVF Conference on Computer Vision and Pattern Recognition (pp. 983-992).
>
> [2] Lukasik, J., Jung, S., & Keuper, M. (2022, October). Learning where to look–generative nas is surprisingly efficient. In European Conference on Computer Vision (pp. 257-273). Cham: Springer Nature Switzerland.
>
> [3] An, S., Lee, H., Jo, J., Lee, S., & Hwang, S. J. (2023). DiffusionNAG: Task-guided Neural Architecture Generation with Diffusion Models. arXiv preprint arXiv:2305.16943.
>
> [4] Ho, J., & Salimans, T. (2022). Classifier-free diffusion guidance. arXiv preprint arXiv:2207.12598.
>
> [5] Ho, J., Jain, A., & Abbeel, P. (2020). Denoising diffusion probabilistic models. Advances in neural information processing systems, 33, 6840-6851.
>
> [6] Muhan Zhang, Shali Jiang, Zhicheng Cui, Roman Garnett, and Yixin Chen. D-VAE: A variational autoencoder for directed acyclic graphs. Advances in Neural Information Processing Systems, 32, 2019

---

### Author Response · Authors · 2024-01-22
**Thank you**

Dear Reviewers and Action Editor,

 We would like to express our sincere thanks to the Action Editor and the reviewers kwHT, uVxU,and 3zRw for their valuable time and effort in providing insightful and helpful suggestions. Their comments helped us to improve the quality of our work.

 The reviewers, along with the encouraging positive comments, raised some excellent questions and concerns, which we have addressed in our responses and in our manuscript. In our updated manuscript, we have highlighted the added text with blue colour and removed text with red colour to enhance readability.

We would also like to inform the reviewers and the Action Editor that are updated manuscript exceeds the limit of 12 pages for 'regular submission' and is now a 'long submission'.

We thank the Action Editor and the reviewers once again for their time and effort in reading and analysing this manuscript.

Sincerely,
DiNAS Authors

---

### Author Response · Authors · 2024-03-09
**Camera-Ready Version**

Dear Action Editor and Reviewers,

The camera-ready version is now uploaded. We would like to thank the reviewers and the action editor again for their input in this paper.

Sincerely,
DiNAS Authors

---

### Decision · Action_Editor_foSU · 2024-02-26

**Recommendation:** Accept as is

**Comment:**

The reviewers were unanimous that this paper should be accepted as is by TMLR, congratulations!

The original reviews outlined many suggestions and questions. The authors replied to almost all of them (very quickly) and provided new comparisons and ablation studies. The authors also clarified several important aspects of the empirical results.

The reviewers' recommendations did not surface new elements.

**Audience:**

Yes, the reviewers are unanimous that this would interest the TMLR audience, especially those interested in the problem of neural architecture search and, possibly, researchers interested in diffusion methods for non-Euclidean domains.

**Claims And Evidence:**

There were some initial concerns from the reviewers regarding the exact claims made in the submission. These were resolved by the authors' response and updates to the submission.